# Neural tangent kernels, transportation mappings, and universal approximation

**Ziwei Ji, Matus Telgarsky, Ruicheng Xian**
Department of Computer Science
University of Illinois at Urbana-Champaign
{ziweiji2,mjt,rxian2}@illinois.edu

## Abstract

This paper establishes rates of universal approximation for the shallow *neural tangent kernel (NTK)*: network weights are only allowed microscopic changes from random initialization, which entails that activations are mostly unchanged, and the network is nearly equivalent to its linearization. Concretely, the paper has two main contributions: a generic scheme to approximate functions with the NTK by sampling from *transport mappings* between the initial weights and their desired values, and the construction of transport mappings via Fourier transforms. Regarding the first contribution, the proof scheme provides another perspective on how the NTK regime arises from rescaling: redundancy in the weights due to resampling allows individual weights to be scaled down. Regarding the second contribution, the most notable transport mapping asserts that roughly $1/\delta^{10d}$ nodes are sufficient to approximate continuous functions, where $\delta$ depends on the continuity properties of the target function. By contrast, nearly the same proof yields a bound of $1/\delta^{2d}$ for shallow ReLU networks; this gap suggests a tantalizing direction for future work, separating shallow ReLU networks and their linearization.

## 1 Main result and overview

Consider functions computed by a single ReLU layer, meaning

$$x \mapsto \sum_{j=1}^{m} s_j \sigma\left(\langle w_j, x \rangle + b_j\right), \tag{1.1}$$

where $\sigma(z) := \max\{0, z\}$. While shallow networks are celebrated as being *universal approximators* (Cybenko, 1989; Funahashi, 1989; Hornik et al., 1989) — they approximate continuous functions arbitrarily well over compact sets — what is more shocking is that gradient descent can learn the parameters to these networks, and they generalize (Zhang et al., 2016).

Working towards an understanding of gradient descent on shallow (and deep!) networks, researchers began investigating the *neural tangent kernel (NTK)* (Jacot et al., 2018; Du et al., 2018; Allen-Zhu et al., 2018), which replaces a network with its linearization at initialization, meaning

$$x \mapsto \frac{\epsilon}{\sqrt{m}} \sum_{j=1}^{m} s_j \langle \tau_j, \tilde{x} \rangle \sigma'\left(\langle \tilde{w}_j, \tilde{x} \rangle\right), \quad \text{where } \tilde{x} = (x, 1) \in \mathbb{R}^{d+1}, \ \tilde{w} = (w, b) \in \mathbb{R}^{d+1}; \tag{1.2}$$

here each $\tilde{w}_j = (w_j, b_j)$ is frozen at Gaussian initialization (henceforth the bias is collapsed in for convenience), and each *transported weight* $\tau_j$ is microscopically close to the corresponding initial weight $\tilde{w}_j$, concretely $\|\tau_j - \tilde{w}_j\| = \mathcal{O}(1/\epsilon\sqrt{m})$, where $\epsilon > 0$ is a parameter and the scaling $\epsilon/\sqrt{m}$ is conventional in this literature (Allen-Zhu et al., 2018).

As eq. (1.2) is merely *affine* in the parameters, it is not outlandish that gradient descent can be analyzed. What *is* outlandish is firstly that gradient descent on eq. (1.1) with small learning rate will track the behavior of eq. (1.2), and secondly the weights hardly change *as a function of $m$*, specifically $\|\tau_j - \tilde{w}_j\|_2 = \mathcal{O}(1/\epsilon\sqrt{m})$.

**Contributions.** This work provides rates of function approximation for the NTK as defined in eq. (1.2), moreover in the "NTK setting": the transported weights must be near initialization, meaning $\|\tau_j - \tilde{w}_j\| = \widetilde{\mathcal{O}}(1/\epsilon\sqrt{m})$. In more detail:

**Continuous functions (cf. Theorem 1.5).** The main theorem packages the primary tools here to say: the NTK can approximate continuous functions arbitrarily well, so long as the width is at least $1/\delta^{10d}$, where $\delta$ depends on continuity properties of the target function; moreover, the transports satisfy $\|\tau_j - \tilde{w}_j\| = \mathcal{O}(1/\epsilon\sqrt{m})$, and the ReLU network (eq. (1.1)) and its linearization (eq. (1.2)) stay close. Re-using many parts of the proof, a nearly-optimal rate $1/\delta^{2d}$ is exhibited for ReLU networks in Theorem E.1; this gap between ReLU networks and their NTK poses a tantalizing gap for future work.

**Approximation via sampling of transport mappings.** The first component of the proof of Theorem 1.5, detailed in Section 2, is a procedure which starts with an infinite width network, and describes how sampling introduces redundancy in the weights, and automatically leads to the desired microscopic transports $\|\tau_j - \tilde{w}_j\| = \mathcal{O}(1/\epsilon\sqrt{m})$. As detailed in Theorem 2.1, the error between the infinite width and sampled networks is $\widetilde{\mathcal{O}}(\epsilon + 1/\sqrt{m})$. In this way, the analysis provides another perspective on the scaling behavior and small weight changes of the NTK.

**Construction of transport mappings.** The second component of the proof of Theorem 1.5, detailed in Section 3, is to construct explicit transport mappings for various types of functions. As detailed in Lemma 3.3, approximating continuous functions proceeds by constructing an infinite width network not directly for the target function $f$, but instead its convolution $f * G_\alpha$ with a Gaussian $G_\alpha$ with tiny variance $\alpha^2$. Care is needed in order to obtain a rate of the form $1/\delta^{\mathcal{O}(d)}$, rather than, say, $1/\delta^{\mathcal{O}(d/\delta)}$. The main constructions are based on Fourier transforms.

Rounding out the organization of this paper: this introduction will state the main summarizing result and its intuition, and then close with related work; Section 4 will describe certain odds and ends for approximating continuous functions which were left out from the main tools in Section 2 and Section 3; Section 5 sketches abstract approaches to constructing transport mappings, including ones based on a corresponding Hilbert space; Section 6 will conclude with open problems and related discussion. Proofs are sketched in the paper body, but details are deferred to the appendices.

## 1.1 BASIC NOTATION, INTUITION, AND MAIN RESULT

The NTK views networks as finite width realizations of intrinsically *infinite width* objects. In order to transport an infinite number of parameters away from their initialization, one option is to use a *transport mapping* $\mathcal{T} : \mathbb{R}^{d+1} \to \mathbb{R}^{d+1}$ to show where weights should go:

$$x \mapsto \mathbb{E}_{\tilde{w}} \left\langle \mathcal{T}(\tilde{w}), \Phi(x; \tilde{w}) \right\rangle = \mathbb{E}_{\tilde{w}} \left\langle \mathcal{T}(\tilde{w}), \tilde{x} \right\rangle \sigma'(\langle \tilde{w}, \tilde{x} \rangle) = \mathbb{E}_{\tilde{w}} \left\langle \mathcal{T}(\tilde{w}), \tilde{x} \right\rangle \mathbb{1} \left[ \langle \tilde{w}, \tilde{x} \rangle \geq 0 \right],$$

where $\Phi(x; \tilde{w}) = \tilde{x}\sigma'(\langle \tilde{w}, \tilde{x} \rangle)$ is a *random feature* representation of $x$ (Rahimi & Recht, 2008). This abstracts the individual transported weights $(\tau_j)_{j=1}^m$ from before into transported weights defined over arbitrary weights $\tilde{w} \in \mathbb{R}^{d+1}$. These (augmented) weights $\tilde{w} = (w, b)$ (with weight $w \in \mathbb{R}^d$ and bias $b \in \mathbb{R}$) will always be distributed according to a standard Gaussian with identity covariance, with $G$ denoting the density and probability law simultaneously.

A key message of this work, developed in Section 2, is (a) the infinite width network can be sampled to give rates of approximation by finite width networks, (b) the microscopic adjustments of the NTK setting arise naturally from the sampling process! Indeed, letting $s \in \{-1, +1\}$ denote a uniformly distributed random sign,

$$\mathbb{E}_{\tilde{w}} \left\langle \mathcal{T}(\tilde{w}), \Phi(x; \tilde{w}) \right\rangle = \mathbb{E}_{\tilde{w},s} \left\langle s^2 \mathcal{T}(\tilde{w}) + s\tilde{w}\boldsymbol{\epsilon}\sqrt{\boldsymbol{m}}, \Phi(x; \tilde{w}) \right\rangle \qquad \because \mathbb{E}\, s^2 = 1, \mathbb{E}\, s = 0$$

$$\approx \frac{1}{m} \sum_{j=1}^m \left\langle s_j^2 \mathcal{T}(\tilde{w}_j) + s_j \tilde{w}_j \boldsymbol{\epsilon}\sqrt{\boldsymbol{m}}, \Phi(x; \tilde{w}_j) \right\rangle \qquad \text{sampling } (w_j, s_j) \tag{1.3}$$

$$= \frac{\boldsymbol{\epsilon}}{\sqrt{\boldsymbol{m}}} \sum_{j=1}^m \left\langle \frac{s_j \mathcal{T}(\tilde{w}_j)}{\boldsymbol{\epsilon}\sqrt{\boldsymbol{m}}} + \tilde{w}_j, s_j \Phi(x; \tilde{w}_j) \right\rangle.$$

As highlighted by the bolded terms: increasing the width $m$ corresponds to resampling, and allows the transported weights to be scaled down! Indeed, the distance moved is $\mathcal{O}(1/\epsilon\sqrt{m})$ *by construction*. To this end, for convenience define

$$
\begin{aligned}
\tau_j &:= \mathcal{T}_\epsilon(\tilde{w}_j, s_j) := \frac{s_j \mathcal{T}(\tilde{w}_j)}{\boldsymbol{\epsilon\sqrt{m}}} + \tilde{w}_j \mathbb{1}[\|\tilde{w}_j\| \le R], \\
\phi_j(x) &:= \Phi_\epsilon(x; \tilde{w}_j, s_j) := \frac{\boldsymbol{\epsilon s_j}}{\boldsymbol{\sqrt{m}}} \Phi(x; \tilde{w}_j) = \frac{\boldsymbol{\epsilon s_j}}{\boldsymbol{\sqrt{m}}} \tilde{x}\sigma'(\langle \tilde{w}_j, \tilde{x}\rangle),
\end{aligned}
\tag{1.4}
$$

where $R$ is a truncation radius included for purely technical reasons. The transport mappings constructed in Section 3 satisfy $B := \sup_{\tilde{w}} \|\mathcal{T}(\tilde{w})\| < \infty$, and thus $\max_j \|\tau_j - \tilde{w}_j\| \le B/\epsilon\sqrt{m}$ by construction as promised (with high probability).

The key message of Section 2 is to control the deviations of this process, culminating in Theorem 2.1 and also Theorem 1.5 below, which yields upper bounds on the width necessary to approximate infinite width networks. The notion of approximation here will follow (Barron, 1993) and use the $L_2(P)$ metric, where $P$ is a probability measure on the ball $\{x \in \mathbb{R}^d : \|x\| \le 1\}$:

$$
\|h\|_{L_2(P)} = \sqrt{\int h(x)^2 \, \mathrm{d}P(x)}.
$$

Additionally $\|h\|_{L_2} = \sqrt{\int h(x)^2 \, \mathrm{d}x}$ and $\|h\|_{L_1} = \int |h(x)| \, \mathrm{d}x$ will respectively denote the usual $L_2$ and $L_1$ metrics over functions on $\mathbb{R}^d$.

**Theorem 1.5** (Simplification of Theorems 2.1 and 4.3). *Let continuous function $f : \mathbb{R}^d \to \mathbb{R}$ be given, along with $\delta \in (0, 1]$ so that $|f(x) - f(x')| \le \epsilon$ whenever $\|x - x'\|_2 \le \delta$ and $\max\{\|x\|, \|x'\|\} \le 1 + \delta$. Let $P$ be any probability distribution over $\|x\| \le 1$. Then there exists a transport mapping $\mathcal{T}$ (defining $\mathcal{T}_\epsilon$ and $\tau_j$ as in eq. (1.4)) and associated scalars*

$$
B := \sup_{\tilde{w}} \|\mathcal{T}(\tilde{w})\|_2 = \widetilde{\mathcal{O}}\left(\frac{M^5 d^{(5d+9)/2}}{\epsilon^4 \delta^{5(d+1)}}\right), \qquad \text{where } M := \sup_{\|x\| \le 1 + \delta} |f(x)|,
$$

*so that with probability at least $1 - 3\eta$ over Gaussian weights $(\tilde{w}_j)_{j=1}^m$ and uniform signs $(s_j)_{j=1}^m$, then $\max_j \|\tau_j - \tilde{w}_j\| \le B/\epsilon\sqrt{m}$, and*

$$
\left\| f - \sum_{j=1}^m \langle \tau_j, \phi_j(\cdot) \rangle \right\|_{L^2(P)} \le \widetilde{\mathcal{O}}\left(\left[\frac{B}{\sqrt{m}} + \epsilon\sqrt{d}\right]\sqrt{\ln(1/\eta)}\right),
$$

$$
\left\| \frac{\epsilon}{\sqrt{m}} \sum_{j=1}^m s_j \sigma(\langle \tau_j, \tilde{x}\rangle) - \sum_{j=1}^m \langle \tau_j, \phi_j(\cdot) \rangle \right\|_{L_2(P)} \le \widetilde{\mathcal{O}}\left(\left[\frac{B^2}{\epsilon m^{3/2}} + \frac{B\sqrt{d}}{m} + \frac{B}{\sqrt{m}} + \epsilon\sqrt{d}\right]\sqrt{\ln(1/\eta)}\right).
$$

In words: given an arbitrary target function $f$ and associated continuity parameter $\delta$, width $(B/\epsilon)^2 = \widetilde{\mathcal{O}}(d^{5d+9}/\epsilon^{10}\delta^{10(d+1)})$ suffices for error $\widetilde{\mathcal{O}}(\epsilon)$, parameters are close to initialization, and the NTK and the original network behave similarly. The randomized construction does not merely give existence, but holds with high probability: the sampling process is thus in a sense robust, and may be used algorithmically!

As provided in Theorem 4.5, elements of the proof of Theorem 1.5 can be extracted and converted into a direct approximation rate of continuous functions by ReLU networks, and the rate becomes $\widetilde{\mathcal{O}}(d^{d+2}/\epsilon^2\delta^{2d+2})$. Since this rate is nearly tight, together these rates pose an interesting question: is there a purely approximation-theoretic gap between shallow ReLU networks and their NTK?

## 1.2 RELATED WORK

**Optimization literature; the NTK.** This work is motivated and inspired by the optimization literature, which introduced the NTK to study gradient descent in a variety of nearly-parallel works (Jacot et al., 2018; Du et al., 2018; Du et al., 2018; Allen-Zhu et al., 2018; Arora et al., 2019; Oymak

& Soltanolkotabi, 2019; Li & Liang, 2018; Cao & Gu, 2019). These works require the network width to be polynomial in $n$, the size of the training set; by contrast, the analysis here studies closeness in function space, and the width instead scales with properties of the target function.

One close relative to the present work is that of (Chizat & Bach, 2019), which provides an abstract proof scheme following the preceding works, and explains the microscopic change of the weights as a consequence of the scaling $\epsilon/\sqrt{m}$. This is consistent with the resampling perspective here, as summarized in eq. (1.3).

**Random features and the mean-field perspective.** The *random features* perspective (Rahimi & Recht, 2008) studies a related *convex* problem: similarly to the NTK, the activations $\sigma'(\langle \tilde{w}_j, \tilde{x} \rangle)$ are held fixed, and what are trained are scalar weights $a_j \in \mathbb{R}$ on each feature. The Fourier transport map construction used both for the NTK here in Theorem 1.5 and for shallow networks in Theorem E.1 proceeds by constructing exactly such a reweighting, and thus the present work also establishes universal approximation properties of random features. A related perspective is presented in the mean-field literature, which relate gradient descent on $(\tilde{w}_j)_{j=1}^m$ to a Wasserstein flow in the space of distributions on these features (Chizat & Bach; Mei et al., 2018). The analysis here does not have any explicit ties to the mean-field literature, however it is interesting and suggestive that transport mappings appear in both.

**Approximation literature.** The closest prior work is due to Barron (1993), who gave good rates of approximation for functions $f : \mathbb{R}^d \to \mathbb{R}$ when the associated quantity $\int \|w\| \cdot |\hat{f}(w)| \, \mathrm{d}w$ is small, where $\hat{f}$ denotes the Fourier transform of $f$. The proofs in Section 3 will use elements from the proofs in (Barron, 1993), but with many distinct components, and thus it is interesting that the same quantity $\int \|w\| \cdot |\hat{f}(w)| \, \mathrm{d}w$ arises once again. Like the work of (Barron, 1993), the present work also chooses to approximate in the $L_2(P)$ metric. Standard classical works in this literature are general universal approximation guarantees without rates or attention to the weight magnitudes (Cybenko, 1989; Hornik et al., 1989; Funahashi, 1989; Leshno et al., 1993). The rate given here of roughly $1/\delta^{2d}$ in Theorem 4.5 does not seem to appear rigorously in prior work, though it is mentioned as a consequence of a proof in (Mhaskar & Micchelli, 1992), who also take the approach of approximation via Gaussian convolutions; the use of convolutions is not only standard (Wendland, 2004), it is moreover classical, having been used in Weierstrass's original proof (Weierstrass, 1885).

Many related works use a RKHSes directly. Sun et al. (2018) prove universal approximation (with rates) via an RKHS, however they do not consider the NTK (or the NTK setting of small weight changes). Bach (2017a) (see also (Bach, 2017b; Basri et al., 2019; Bietti & Mairal, 2019)) studies a variety of questions related to function fitting with the random features model, including establishing rates of approximation for Lipschitz functions on the surface of the sphere (with a few further conditions); the rates are better than those here (roughly $\Theta(1/\delta^{d/2})$), however they do not consider the NTK setting, meaning either the setting of small changes from initialization nor the linearization.

Another close parallel work studies exact representation power of infinite width networks, developing representations for functions with $\Omega(d)$ derivatives (Ongie et al., 2019); similarly, the constructions here use an exact representation result for Gaussian convolutions, as developed in Section 3.

Regarding lower bounds from the literature, there are two lower bounds of the form $1/\delta^{d/2}$ for general shallow networks, not necessarily in the NTK setting (Yarotsky, 2016; Bach, 2017a). Interestingly, Yarotsky (2016) also presents a lower bound of $1/\delta^d$ for approximations whose parameters vary *continuously* with the target function; this seems to hold for the Fourier constructions here in Section 3, though an argument needs to be made for the sampling step.

## 2 SAMPLING FROM A TRANSPORT

This section establishes that by sampling from an infinite width NTK, the resulting finite width NTK is close in $L_2(P)$ both to the infinite width idealization, and also to the finite width non-linearized ReLU network; moreover, the sampling process introduces redundancy in the weights, allowing them to be scaled down and lie close to initialization.

**Theorem 2.1.** *Suppose $B \geq \max\left\{2, \sup_{\tilde{w}} \|\mathcal{T}(\tilde{w})\|_2\right\}$, and set $R := \sqrt{d+1} + 2\sqrt{\ln(m/\eta)}$. With probability at least $1 - 3\eta$, then $\max_j \|\tau_j - \tilde{w}_j\| \leq {}^B\!/\!_{\epsilon\sqrt{m}}$, and*

$$\left\| \sum_{j=1}^m \left\langle \tau_j, \phi_j(\cdot) \right\rangle - \mathbb{E}_{\tilde{w}} \left\langle \mathcal{T}(\tilde{w}), \Phi(\cdot; \tilde{w}) \right\rangle \right\|_{L^2(P)} \leq 2 \left( \frac{B}{\sqrt{m}} + \epsilon R \right) \left[ 1 + \sqrt{\ln(1/\eta)} \right], \tag{2.2}$$

$$\left\| \sum_j \left\langle \tau_j, \phi_j(\cdot) \right\rangle - \sum_j \frac{s_j \epsilon}{\sqrt{m}} \sigma(\left\langle \tau_j, \cdot \right\rangle) \right\|_{L_2(P)} \leq 2 \left( \frac{B^2}{\epsilon m^{3/2}} + \frac{BR}{m} + \frac{B}{\sqrt{m}} + \epsilon R \right) \left[ 1 + \sqrt{\ln(1/\eta)} \right]. \tag{2.3}$$

As discussed in the introduction, $\max_j \|\tau_j - \tilde{w}_j\| \leq {}^B\!/\!_{\epsilon\sqrt{m}}$ is essentially by construction. Next, recall the sampling derivation in eq. (1.3), restated here as a lemma for convenience, the notation $(\mathcal{W}, S)$ collecting all random variables together, meaning $\mathcal{W} = (\tilde{w}_1, \ldots, \tilde{w}_m)$ and $S = (s_1, \ldots, s_m)$.

**Lemma 2.4.** $\mathbb{E}_{\tilde{w}} \left\langle \mathcal{T}(\tilde{w}), \Phi(x; \tilde{w}) \right\rangle = \mathbb{E}_{\widetilde{W}, S} \sum_j \left\langle \mathcal{T}_\epsilon(\tilde{w}_j, s_j), \Phi_\epsilon(x; \tilde{w}_j, s_j) \right\rangle = \mathbb{E}_{\widetilde{W}, S} \sum_j \left\langle \tau_j, \phi_j(x) \right\rangle.$

The proof of eq. (2.2) now follows from the classical Maurey sampling lemma (Pisier, 1980), which was also used in the related work by Barron (1993). The following version additionally includes a high probability control, which results from an application of McDiarmid's inequality. Applying the following sampling lemma to the present setting, the deviations will scale with $B := \sup_{\tilde{w}} \|\mathcal{T}(\tilde{w})\|_2$.

**Lemma 2.5** (Maurey). *Let functions $\{g(\cdot; v) : v \in \mathcal{V}\}$ be given, where $\mathcal{V} \subseteq \mathbb{R}^p$ is a set of possible parameters. Let $\nu$ be a probability measure over $\mathcal{V}$, let $(v_1, \ldots, v_m)$ be an iid random draw from $\nu$, and define*

$$f(x) := \mathbb{E}_{v \sim \nu} g(x; v) \qquad \text{and} \qquad g_j(x) := g(x; v_j).$$

*Then*

$$\mathbb{E}_{((s_j, v_j))_{j=1}^m} \left\| f - \frac{1}{m} \sum_{j=1}^m g_j \right\|_{L^2(P)}^2 \leq \frac{1}{m} \mathbb{E}_v \left\| g(\cdot; v) \right\|_{L_2(P)}^2 \leq \frac{1}{m} \sup_{v \in \mathcal{V}} \left\| g(\cdot; v) \right\|_{L_2(P)}^2,$$

*and with probability at least $1 - \eta$,*

$$\left\| f - \frac{1}{m} \sum_{j=1}^m g_j \right\|_{L^2(P)} \leq \sup_{v \in \mathcal{V}} \|g(\cdot; v)\|_{L_2(P)} \left[ \frac{1 + \sqrt{2 \ln(1/\eta)}}{\sqrt{m}} \right].$$

Concretely, here $g_j(x) = m \left\langle \tau_j, \phi_j(x) \right\rangle$, and $\sup_{v \in \mathcal{V}} \|g(\cdot; \tilde{w})\|_{L_2(P)} \leq \sqrt{2} \sup_{\tilde{w}} \|\mathcal{T}(\tilde{w})\|_2 = \mathcal{O}(B + R\epsilon\sqrt{m})$ by Cauchy-Schwarz. Before continuing, note also that there are other proof schemes attaining similar bounds (Bach, 2017a;b, Proposition 1), and that similar bounds are possible for the uniform norm, albeit with more sensitivity to the basis functions $g$ (cf. Lemma B.2).

Turning now to the final bound in eq. (2.3), the first step is to note by positive homogeneity of the ReLU that $\sigma(\left\langle \tau_j, \tilde{x} \right\rangle) = \left\langle \tau_j, \tilde{x} \right\rangle \sigma'(\left\langle \tau_j, \tilde{x} \right\rangle)$, thus

$$\sum_{j=1}^m \left\langle \tau_j, \phi_j \right\rangle - \frac{\epsilon}{\sqrt{m}} \sum_{j=1}^m s_j \sigma(\left\langle \tau_j, \tilde{x} \right\rangle) = \sum_{j=1}^m \left\langle \tau_j, \phi_j - \frac{s_j \epsilon}{\sqrt{m}} \tilde{x} \sigma'(\left\langle \tau_j, \tilde{x} \right\rangle) \right\rangle,$$

which boils down to checking the difference in activations, namely $\sigma'(\left\langle \tilde{w}_j, \tilde{x} \right\rangle) - \sigma'(\left\langle \tau_j, \tilde{x} \right\rangle)$. As is standard in the NTK literature, since $\tau_j - \tilde{w}_j$ is (with high probability) microscopic compared to $\left\langle \tilde{w}_j, \tilde{x} \right\rangle$, the activations should also be close. The following lemma makes this precise.

**Lemma 2.6.** *For any $x \in \mathbb{R}^p$, if $R \geq \sqrt{d} + 2\sqrt{\ln\left(\frac{\epsilon\sqrt{m\pi}}{B\sqrt{2}}\right)}$ (as used in eq. (1.4)), then*

$$\mathbb{E}_{\tilde{w}} \left| \sigma'(\left\langle \tilde{w}, \tilde{x} \right\rangle) - \sigma'(\left\langle \mathcal{T}_\epsilon(\tilde{w}), \tilde{x} \right\rangle) \right| \leq \frac{2B\sqrt{2}}{\epsilon\sqrt{m\pi}}.$$

From here, the eq. (2.3) can be established with another application of Lemma 2.5. This completes the proof of Theorem 2.1 after an application of Gaussian concentration to ensure $\max_j \|\tilde{w}_j\| \leq R$. This also establishes the first half of Theorem 1.5.

## 3 CONCRETE TRANSPORT MAPPINGS VIA FOURIER TRANSFORMS

The previous section showed function approximation in the NTK setting *assuming* the existence of an infinite width NTK defined by a transport mapping $\mathcal{T}$; this section will close the gap by providing a variety of transport maps.

The transport mappings here will be constructed via Fourier transforms, with convention

$$\hat{f}(x) = \int \exp\left(-2\pi i x^\intercal w\right) f(x) \, \mathrm{d}x;$$

a few general properties are summarized in Appendix A. Interestingly, these transports are all *random feature transports*: they have the form $\mathcal{T}(\tilde{w}) = (0, \cdots, 0, p(\tilde{w}))$ where $p$ is a *signed density* over random features, and $\mathbb{E}_{\tilde{w}} \langle \mathcal{T}(\tilde{w}), \Phi(x; \tilde{w}) \rangle = \mathbb{E}_{\tilde{w}} p(\tilde{w}) \sigma'(\langle \tilde{w}, \tilde{x} \rangle)$. This perspective of a signed density will be used to prove universal approximation — again via sampling! — of shallow ReLU networks (and random features) later in Theorems 4.5 and E.1. (For constructions which are not based on random features, see Section 5.)

The first steps of the approach here follow a derivation due to Barron (1993). Specifically, the *inverse* Fourier transform gives a way to rewrite a function as an infinite with network with complex-valued activations $x \mapsto \exp(2\pi i x^\intercal w)$:

$$f(x) = \int \exp(2\pi i x^\intercal w) \hat{f}(w) \, \mathrm{d}w.$$

A key trick due to Barron (1993) is to force the right hand side to be real (since the left hand side is real): specifically, letting $\hat{f}(w) = |\hat{f}(w)| \exp(2\pi i \theta_f(w))$ with $|\theta_f(w)| \leq 1$ denote the radial decomposition of $\hat{f}$,

$$\begin{aligned}
\mathrm{Re}\, f(x) &= \mathrm{Re} \int \exp(2\pi i x^\intercal w) \hat{f}(w) \, \mathrm{d}w \\
&= \mathrm{Re} \int \exp(2\pi i x^\intercal w + 2\pi i \theta_f(w)) |\hat{f}(w)| \, \mathrm{d}w \\
&= \int \cos\left(2\pi(x^\intercal w + \theta_f(w))\right) |\hat{f}(w)| \, \mathrm{d}w.
\end{aligned}$$

After this step, the proofs diverge: the approach here is to use the fundamental theorem of calculus to rewrite $\cos$ in terms of $\sigma'$:

$$\cos(z) - \cos(0) = -\int_0^z \sin(b) \, \mathrm{d}b = -\int_0^\infty \sin(b) \mathbb{1}\left[z - b \geq 0\right] \mathrm{d}b, = -\int_0^\infty \sin(b) \sigma'(z - b) \, \mathrm{d}b;$$

plugging this back in gives an explicit representation of $f$ in terms of an infinite width threshold network! A similar approach can be used to obtain an infinite width ReLU network.

This is summarized in the following lemma, which includes a calculation of the error incurred by truncating the weights; this truncation is necessary when applying the sampling of Section 2. Interestingly, this truncation procedure leads to the quantity $\int \|w\| \cdot |\hat{f}(w)| \, \mathrm{d}w$, which was *explicitly* introduced as a key quantity by Barron (1993) via a different route, namely of introducing a factor $\|w\|$ to enforce decay on $\cos$.

**Lemma 3.1.** *Let $f : \mathbb{R}^d \to \mathbb{R}$ be given with Fourier transform $\hat{f}$ and truncation radius $r \in [0, \infty]$.*

  1. *Define infinite width threshold network*

$$\begin{aligned}
F_r(x) :=\ & f(0) + \int |\hat{f}(w)| \cos\left(2\pi(\theta_f(w) - \|w\|)\right) \, \mathrm{d}w \\
& + 2\pi \int \sigma'(\langle \tilde{w}, \tilde{x} \rangle) |\hat{f}(w)| \sin(2\pi(\theta_f(w) - b)) \mathbb{1}\left[|b| \leq \|w\| \leq r\right] \mathrm{d}\tilde{w}.
\end{aligned}$$

  *For any $\|x\| \leq 1$, $F_\infty = f$ and $\left| f(x) - F_r(x) \right| \leq 4\pi \int_{\|w\| > r} \|w\| \cdot |\hat{f}(w)| \, \mathrm{d}w.$*

2. *Define infinite width ReLU network*

$$Q_r(x) := f(0) + \int |\hat{f}(w)| \left[\cos(2\pi(\theta_f(w) - \|w\|)) - 2\pi\|w\|\sin(2\pi(\theta_f(w) - \|w\|))\right] dw$$

$$+ x^\intercal \int w|\hat{f}(w)| \, dw$$

$$+ 4\pi^2 \int \sigma(\tilde{w}^\intercal \tilde{x})|\hat{f}(w)|\cos(2\pi(\|w\| - b))\mathbb{1}[|b| \le \|w\| \le r] \, d\tilde{w}.$$

*For any* $\|x\| \le 1$, $Q_\infty = f$ *and* $\left|f(x) - Q_r(x)\right| \le 12\pi^2 \int_{\|w\|>r} \|w\|^2 \cdot |\hat{f}(w)| \, dw$.

(The second part of the shows that the same technique allows functions to be written with *equality* as ReLU networks; this is included as a curiosity and used in a few places in the appendices, but is not part of the main NTK story.)

The preceding constructions immediately yield transport mappings from Gaussian initialization to the function $f$ in a brute-force way: by introducing the fraction $G(\tilde{w})/G(\tilde{w})$, calling the numerator part of the integration measure, and the denominator part of the integrand. As stated before, these transport maps are random feature maps: they zero out the coordinates corresponding to $x$!

**Lemma 3.2.** *Let* $f : \mathbb{R}^d \to \mathbb{R}$ *be given with Fourier transform* $\hat{f}$. *For any* $r \in [0, \infty]$, *define transport mapping* $\mathcal{T}_r(w, b) := (0, \ldots, 0, p_r(\tilde{w}))$ *with*

$$\mathcal{T}_r(w, b)_{d+1} = p_r(\tilde{w}) := 2\left[f(0) + \int |\hat{f}(v)|\cos(2\pi(\theta_f(v) - \|v\|)) \, dv\right]$$

$$+ 2\pi\left(\frac{|\hat{f}(w)|}{G(\tilde{w})}\right)\cos(2\pi(\theta_f(w) - b))\mathbb{1}[|b| \le \|w\| \le r].$$

*By this choice, for any* $\|x\| \le 1$, $f(x) = \mathbb{E}_{\tilde{w}} \left\langle \mathcal{T}_\infty(\tilde{w}), \Phi(x; \tilde{w})\right\rangle$, *and*

$$\sup_{\tilde{w}} \|\mathcal{T}_r(\tilde{w})\|_2 \le 2|f(0)| + 2\int |\hat{f}(v)| \, dv + 2\pi \sup_{\substack{\|w\| \le r \\ |b| \le \|w\|}} \frac{|\hat{f}(w)|}{G(\tilde{w})},$$

$$\left|f(x) - \mathbb{E}\left\langle \mathcal{T}(\tilde{w}), \Phi(x; \tilde{w})\right\rangle\right| \le 4\pi \int_{\|w\|>r} |\hat{f}(w)| \cdot \|w\| \, dw.$$

The preceding construction may seem general, however it is quite loose, noting the final supremum term within $\sup_{\tilde{w}} \|\mathcal{T}_r(\tilde{w})\|_2$; indeed, attempting to plug this construction into Theorem 2.1 does not yield the $1/\delta^{\mathcal{O}(d)}$ rate in Theorem 1.5, but instead a rate $1/\delta^{\mathcal{O}(d/\delta)}$, which is disastrously larger!

Interestingly, a fix is possible for special functions of the form $f * G_\alpha$, namely convolutions with Gaussians of coordinate-wise variance $\alpha^2$. These are exactly the types of functions used in Section 4 to approximate continuous functions. The fix is simply to apply a change of variable so that, in a sense, the target function and the initialization distribution have similar units.

**Lemma 3.3.** *Let function* $f$, *variance* $\alpha^2 > 0$, *and* $r \in [0, \infty]$ *be given, and define* $f_\alpha := f * G_\alpha$ *and* $\phi := (2\pi\alpha)^{-1}$, *and transport mapping* $\mathcal{T}_r(w, b) := (0, \ldots, 0, p_r(\tilde{w}))$ *with*

$$\mathcal{T}_r(w, b)_{d+1} = p_r(\tilde{w}) := 2\left[f_\alpha(0) + \int |\hat{f}_\alpha(v)|\cos(2\pi(\theta_{f_\alpha}(v) - \|v\|)) \, dv\right]$$

$$+ 2\pi(2\pi\phi^2)^{(d+1)/2}|\hat{f}(\phi w)|e^{b^2/2}\sin(2\pi(\theta_{f_\alpha}(\phi w) - b))\mathbb{1}[|b| \le \|w\| \le r].$$

*Then* $f_\alpha(x) = \mathbb{E}_{\tilde{w}}\left\langle \mathcal{T}_\infty(\tilde{w}), \Phi(x; \tilde{w})\right\rangle$ *for* $\|x\| \le 1$, *and for* $r \in [\sqrt{d}, \infty)$,

$$\sup_{\tilde{w}} \|\mathcal{T}_r(\tilde{w})\| \le 2\left[M + (2\pi\phi^2)^{d/2}M_f\left(1 + \sqrt{2\pi^3\phi^2}e^{r^2/2}\right)\right],$$

*where* $M := \sup_x |f(x)|$, *and* $M_f = 1$ *when* $f_\alpha = G_\alpha$ *and* $M_f = \|f(\phi\cdot)\|_{L_1}$ *otherwise, and*

$$\sup_{\|x\| \le 1}\left|f(x) - \mathbb{E}\left\langle \mathcal{T}_r(\tilde{w}), \Phi(x; \tilde{w})\right\rangle\right| \le 4\pi(2\pi\phi^2)^{(d+1)/2}M_f(\sqrt{d} + 3)\exp\left(-(r - \sqrt{d})^2/4\right).$$

## 4 APPROXIMATING CONTINUOUS FUNCTIONS

The final piece needed to prove Theorem 1.5 is to show that a function $f$ is close to its Gaussian convolution $f * G_\alpha$, at least when $\alpha > 0$ is chosen appropriately. This is a classical topic (Wendland, 2004), and indeed it was used in the original proof of the Weierstrass approximation theorem (Weierstrass, 1885). The treatment here will include enough detail necessary to yield explicit rates.

The following definition will be used to replace the usual $(\epsilon, \delta)$ conditions associated with continuous functions with an exact quantity.

**Definition 4.1.** Let $f : \mathbb{R}^d \to \mathbb{R}$ be given, and define *modulus of continuity* $\omega_f$ as

$$\omega_f(\delta) := \sup \left\{ f(x) - f(x') : \max\{\|x\|, \|x'\|\} \leq 1 + \delta, \|x - x'\| \leq \delta \right\}. \qquad \diamond$$

If $f$ is continuous, then $\omega_f$ (defined here over a compact set) is not only finite for all inputs, but moreover $\lim_{\delta \to 0} \omega_f(\delta) \to 0$. It is also possible to use this definition with discontinuous functions; note additionally that the convolution bounds in Section 3 only required an $L_1$ bound on the pre-convolution function $f$, and therefore the tools throughout may be applied to discontinuous functions, albeit with some care to their Fourier transforms!

**Lemma 4.2.** *Let $f : \mathbb{R}^d \to \mathbb{R}$ and $\delta > 0$ be given, and define*

$$M := \sup_{\|x\| \leq 1 + \delta} |f(x)|, \qquad f_{|\delta}(x) := f(x)\mathbb{1}[\|x\| \leq 1 + \delta], \qquad \alpha := \frac{\delta}{\sqrt{d} + \sqrt{2 \ln(2M/\omega_f(\delta))}}.$$

*Let $G_\alpha$ denote a Gaussian with the preceding variance $\alpha^2$. Then*

$$\sup_{\|x\| \leq 1} \left| f - f_{|\delta} * G_\alpha \right| \leq 2\omega_f(\delta).$$

The proof splits the integrand into two parts: points close to $x$, and points far from it. Points close to $x$ must behave like $f(x)$ due to continuity, whereas points far from $x$ are rare and do not matter due to the Gaussian convolution. The full details are in the appendix.

Lemma 4.2 can be combined with the transport for $f * G_\alpha$ from Section 3 to give a transport for approximating continuous functions.

**Theorem 4.3.** *As in Lemma 4.2, let $f : \mathbb{R}^d \to \mathbb{R}$ and $\delta > 0$ be given, and define*

$$M := \sup_{\|x\| \leq 1 + \delta} |f(x)|, \qquad\qquad f_{|\delta}(x) := f(x)\mathbb{1}[\|x\| \leq 1 + \delta],$$

$$\alpha := \frac{\delta}{\sqrt{d} + \sqrt{2 \ln(2M/\omega_f(\delta))}} = \widetilde{\mathcal{O}}(\delta/\sqrt{d}), \qquad r := \sqrt{d} + 2\sqrt{\ln \left( \frac{4\pi M_f(\sqrt{d} + 3)}{(2\pi\alpha^2)^{(d+1)/2}\omega_f(\delta)} \right)}.$$

*Let $G_\alpha$ denote a Gaussian with the preceding variance $\alpha^2$, and let $\mathcal{T}_r$ denote the truncated Fourier map constructed in Lemma 3.3 for $f_{|\delta} * G_\alpha$, with preceding truncation choice $r$. Then*

$$\sup_{\tilde{w}} \|\mathcal{T}_r(\tilde{w})\| = \widetilde{\mathcal{O}} \left( \|f_{|\delta}\|_{L_1}^5 \left( \frac{\sqrt{d}}{\delta} \right)^{5(d+1)} \left[ \frac{\sqrt{d}}{\omega_f(\delta)} \right]^4 \right),$$

$$\sup_{\|x\| \leq 1} \left| f - \mathbb{E}_{\tilde{w}} \left\langle \mathcal{T}_r(\tilde{w}), \Phi(x; \tilde{w}) \right\rangle \right| \leq \omega_f(\delta).$$

This completes all the pieces needed to prove Theorem 1.5.

*Proof of Theorem 1.5.* Let $f$ be given, and let $\mathcal{T}_r$ denote the transport mapping provided by Theorem 4.3 for $f_{|\delta} * G_\alpha$, whose various parameters match those in the statement of Theorem 1.5. The proof is completed by plugging $\mathcal{T}_r$ into Theorem 2.1, and simplifying by noting that $\epsilon \geq \omega_f(\delta)$ by definition, and $\|f_{|\delta}\|_{L_1} = \mathcal{O}(M)$ since $\delta \leq 1$. ◻

As mentioned earlier, the infinite width network constructed in Lemma 3.1 via inverse Fourier transforms can be used to succinctly prove (via Lemma 2.5 and Lemma 4.2) that threshold and ReLU networks are universal approximators, with a rate vastly improving upon that of Theorem 1.5.

Before stating the result, one more tool is needed: a sampling semantics for signed densities; see also (Bach, 2017a;b) for further development and references.

**Definition 4.4.** A sample from a signed (Lebesgue) density $p : \mathbb{R}^{d+1} \to \mathbb{R}$ with $\|p\|_{L_1} < \infty$ is a pair $(\tilde{w}, s)$ where $\tilde{w}$ is sampled from the probability density $|p|/\|p\|_{L_1}$, and $s := \text{sgn}(p(\tilde{w}))$. Let $\mathbb{E}_p$ denote the corresponding expectation over $(\tilde{w}, s) \sim p$. $\diamond$

This notion of signed sampling also has a corresponding Maurey lemma, and an analogue for the uniform norm; both are provided in Appendix B. The full detailed universal approximation theorems for threshold and ReLU networks are provided in Appendix E; a simplified form for threshold networks alone is as follows. In either case, the proof proceeds by applying signed density sampling bounds (e.g., appropriate generalizations of Lemma 2.5) to the infinite width networks constructed in Lemma 3.1. Curiously, the simplified bound stated here for threshold networks for the uniform norm is only a multiplicative factor $\sqrt{d}$ larger than the $L_2(P)$ bound in Theorem E.1.

**Theorem 4.5** (Simplification of Theorem E.1). *Let $f : \mathbb{R}^d \to \mathbb{R}$ and $\delta > 0$ be given, and define*

$$M := \sup_{\|x\| \leq 1+\delta} |f(x)|, \qquad f_{|\delta}(x) := f(x)\mathbb{1}[\|x\| \leq 1+\delta], \qquad \alpha := \frac{\delta}{\sqrt{d} + \sqrt{2\ln(2M/\omega_f(\delta))}}.$$

*Then there exist $c \in \mathbb{R}$ and $p : \mathbb{R}^{d+1} \to \mathbb{R}$ with*

$$|c| \leq M + \|f_{|\delta}\|_{L_1}(2\pi\alpha^2)^{d/2}, \qquad \text{and} \qquad \|p\|_{L_1} \leq 2\|f_{|\delta}\|_{L_1}\sqrt{\frac{2\pi d}{(2\pi\alpha^2)^{d+1}}},$$

*so that, with probability $\geq 1 - 3\eta$ over $((s_j, \tilde{w}_j))_{j=1}^m$ drawn from $p$ (cf. Definition 4.4),*

$$\sup_{\|x\| \leq 1} \left| f(x) - \left[ c_1 + \frac{\|p\|_{L_1}}{m} \sum_{j=1}^m s_j \sigma'(\langle \tilde{w}_j, x \rangle) \right] \right| \leq 2\omega_f(\delta) + \frac{\|p\|_{L_1}}{\sqrt{m}} \left[ 8\sqrt{d\ln(m)} + \sqrt{\ln(1/\eta)} \right].$$

## 5 ABSTRACT TRANSPORT MAPPINGS, AND AN RKHS

Section 3 provided *concrete* transport mappings via Fourier transforms: it was, for instance, easy to use these constructions to develop approximation rates for continuous functions. These constructions had a major weakness: they were random feature transport mappings, meaning they arguably did not fully utilize the transport sampling provided in Section 2. This section will develop one abstract approach via RKHSes, but first will revisit the random feature constructions.

Suppose $f(x) = \int p(\tilde{w})\sigma'(\langle \tilde{w}, \tilde{x} \rangle) \, d\tilde{w}$ for some density $p$; as in the proof of Lemma 3.2, introducing the term $G(\tilde{w})/G(\tilde{w})$ gives $f(x) = \int \frac{p(\tilde{w})}{G(\tilde{w})}\sigma'(\langle \tilde{w}, \tilde{x} \rangle) \, dG(\tilde{w})$, which is now in the desired form, however the ratio term can be large (and a truncation is needed to make it finite in Lemma 3.2). The refined construction in Lemma 3.3 achieved a better bound on $\sup_{\tilde{w}} \|\mathcal{T}(\tilde{w})\|$ by being careful about the scaling of the Gaussian, and then standardizing it with a change-of-variable transformation, but still it yields a random feature transport.

Another approach would be to start from the second construction in Lemma 3.1, which writes $f(x) = \int p(\tilde{w})\sigma(\langle \tilde{w}, \tilde{x} \rangle) \, d\tilde{w} = \int p(\tilde{w}) \langle \tilde{w}, \tilde{x} \rangle \, \sigma'(\langle \tilde{w}, \tilde{x} \rangle)$, and thus build a transport around $\tilde{w} \mapsto p(\tilde{w})\tilde{w}$, which now uses all coordinates. This transport mapping is still just a rescaling, however, and does not lead to improvements when plugged into the other parts of this work.

Consider the following approach to building a general $\mathcal{T}$ and an associated *Reproducing Kernel Hilbert Space (RKHS)*. To start, define an inner product $\langle \cdot, \cdot \rangle_{\mathcal{H}}$ and norm $\| \cdot \|_{\mathcal{H}}$ via $\|\mathcal{T}\|_{\mathcal{H}}^2 = \langle \mathcal{T}, \mathcal{T} \rangle_{\mathcal{H}} = \int \|\mathcal{T}(\tilde{w})\|_2^2 \, dG(\tilde{w}) = \|\mathcal{T}\|_{L_2(G)}^2$; to justify this Hilbert space, note that it gives rise to the usual kernel product (Cho & Saul, 2009), namely

$$(x, x') \mapsto \mathbb{E}_{\tilde{w}} \Phi(x; \tilde{w})^\top \Phi(x'; \tilde{w}) = \langle \Phi(x; \cdot), \Phi(x'; \cdot) \rangle_{\mathcal{H}},$$

and moreover our earlier predictors can be written as $\mathbb{E}_{\tilde{w}} \langle \mathcal{T}(\tilde{w}), \Phi(x; \tilde{w}) \rangle = \langle \mathcal{T}, \Phi(x; \cdot) \rangle_{\mathcal{H}}$.

The utility of these definitions is highlighted in the following bounds; specifically, while a given $\mathcal{T}$ may have $\sup_{\tilde{w}} \|\mathcal{T}(\tilde{w})\|_2 = \infty$, truncation can make this quantity finite (and thus Theorem 2.1 may be applied), and the approximation error can be bounded with $\|\mathcal{T}\|_{\mathcal{H}}$.

**Proposition 5.1.**     *1. The* output-truncated *transport* $\mathcal{T}_B(\tilde{w}) := \mathcal{T}(\tilde{w})\mathbb{1}\left[\|\mathcal{T}(\tilde{w})\|_2 \leq B\right]$ *has approximation error*

$$\sup_{\|x\| \leq 1} \left| \langle \mathcal{T}_B, \Phi(x; \cdot) \rangle_{\mathcal{H}} - \langle \mathcal{T}, \Phi(x; \cdot) \rangle_{\mathcal{H}} \right| \leq \frac{\|\mathcal{T}\|_{\mathcal{H}}^2 \sqrt{2}}{B^2}.$$

*2. The* input-truncated *transport* $\mathcal{T}_r(\tilde{w}) := \mathcal{T}(\tilde{w})\mathbb{1}[\|\tilde{w}\| \leq r]$ *with* $r > \sqrt{d}$ *has approximation error*

$$\sup_{\|x\| \leq 1} \left| \langle \mathcal{T}_r, \Phi(x; \cdot) \rangle_{\mathcal{H}} - \langle \mathcal{T}, \Phi(x; \cdot) \rangle_{\mathcal{H}} \right| \leq \frac{\|\mathcal{T}\|_{\mathcal{H}} \sqrt{2}}{e^{(r-\sqrt{d})^2/4}}.$$

Unfortunately, this formalism can not be applied to the pre-truncation mapping $\mathcal{T}_\infty$ from Lemma 3.3, since $\|\mathcal{T}_\infty\|_{\mathcal{H}} = \infty$. Consequently, this approach is left as an interesting direction for future work.

## 6 OPEN PROBLEMS

The main open question is: how much can the rates $1/\delta^{2d}$ for ReLU networks and $1/\delta^{10d}$ for their NTK be tightened, and is there a genuine gap? Expanding this inquiry, firstly there are three relevant choices regarding which layers are trained: training just the output layer as with random features (Bach, 2017b;a), training just the input layer (as in this work), and training both layers. Secondly, for each of these choices, there is a question of norm; e.g., by requiring the maximum over node weight Euclidean norms to be small, the NTK regime is enforced. Are there genuine separations between these settings? Which settings are most relevant empirically? What happens beyond the NTK (Allen-Zhu & Li, 2019)?

Another direction is to use the Fourier tools of Section 2, as well as other tools for constructing transportation maps, and identify function classes with good approximation rates by the NTK and by shallow networks, in particular rates with a merely polynomial dependence on dimension.

Connecting back to the optimization literature, the referenced NTK optimization works for the squared loss seem to require a width which scales with $n$, and the test error sometimes scales with detailed functions of the observed labels, which require a further argument to go to $0$ (see, e.g., $y^{\intercal}(H^\infty)^{-1}y$ in (Arora et al., 2019)). Perhaps such quantities can be replaced with a function space or other approximation theoretic perspective on the conditional mean function (and samples thereof)?

Lastly, what are connections to *optimal* transport? It seems natural to choose $\mathcal{T}$ as an optimal transport, in which case one would hope the parameter $B := \sup_{\tilde{w}} \|\mathcal{T}(\tilde{w})\|_2$ can be small, and moreover easily bounded by the optimal transport cost, ideally in ways similarly easy to the bounding by the Hilbert norm in Proposition 5.1.

### ACKNOWLEDGEMENTS

The authors are grateful for support from the NSF under grant IIS-1750051, and from NVIDIA via a GPU grant.

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

## A  TECHNICAL LEMMAS

**Gaussian Concentration.**  The following lemma collects a few properties of Gaussian concentration needed throughout.

**Lemma A.1.** *Let $w \sim G_d$ be a standard Gaussian in $\mathbb{R}^d$, and let $r \geq \sqrt{d}$ be given.*

1. $\int \|w\| \, \mathrm{d}G(w) \leq \sqrt{d}$.

2. $\mathbb{P}[\|w\| > r] \leq \exp(-(r - \sqrt{d})^2/2)$; *alternatively, with probability at least $1 - \eta$, $\|w\| \leq \sqrt{d} + \sqrt{2\ln(1/\eta)}$.*

3. $\int_{\|w\|>r} \|w\| \, \mathrm{d}G(w) \leq (r+2)\exp\left(-(r-\sqrt{d})^2/2\right) \leq 2(\sqrt{d}+3)\exp\left(-(r-\sqrt{d})^2/4\right)$.

4. $\int_{\|w\|>r} \|w\|^2 \, \mathrm{d}G(w) \quad \leq \quad 2(r+2)^2\exp\left(-(r-\sqrt{d})^2/2\right) \quad \leq \quad 2(\sqrt{d}+7)^2\exp\left(-(r-\sqrt{d})^2/4\right)$.

The more convenient form of some of the inequalities will need to following technical lemma.

**Lemma A.2.** *Given $b \geq 0$ and $c > 0$ and $a \geq 0$ with $a + b \geq 2c$ and $x \geq b$, then*

$$(x + a) \exp(-(x - b)^2/c) \leq (a + b) \exp(-(x - b)^2/(2c)),$$

*and if moreover $a + b \geq 4c$,*

$$(x + a)^2 \exp(-(x - b)^2/c) \leq (a + b)^2 \exp(-(x - b)^2/(2c))$$

*Proof.* Since $\ln(x + a) \leq \ln(b + a) + (x - b)/(b + a)$,

$$
\begin{aligned}
(x + a) \exp(-(x - b)^2/c) &\leq (a + b) \exp(-(x - b)^2/c + (x - b)/(b + a)) \\
&\leq (a + b) \exp(-(x - b)^2/c + (x - b)^2/(2c)) \\
&\leq (a + b) \exp(-(x - b)^2/(2c)).
\end{aligned}
$$

Similarly, multiplying the preceding Taylor expansion by 2,

$$
\begin{aligned}
(x + a)^2 \exp(-(x - b)^2/c) &\leq (a + b)^2 \exp(-(x - b)^2/c + 2(x - b)/(b + a)) \\
&\leq (a + b)^2 \exp(-(x - b)^2/c + 2(x - b)^2/(4c)) \\
&\leq (a + b)^2 \exp(-(x - b)^2/(2c)).
\end{aligned}
$$

$\square$

*Proof of Lemma A.1.*  1. By Jensen's inequality, $\int \|w\| \, \mathrm{d}G(w) \leq \sqrt{\int \|w\|^2 \, \mathrm{d}G(w)} = \sqrt{d}$.

2. The claim follows from Gaussian concentration with Lipschitz mappings (Wainwright, 2015, Theorem 2.4), specifically since $w \mapsto \|w\|$ is 1-Lipschitz, meaning

$$\left| \|w\| - \|w'\| \right| \leq \|w - w'\|,$$

and since $\mathbb{E} \|w\| < \sqrt{d}$.

3. Note that

$$\int_{\|w\|>r} \|w\| \, \mathrm{d}G(w) \leq \sum_{i=0}^{\infty} \int_{r+i<\|w\|\leq r+i+1} (r + i + 1) \, \mathrm{d}G(w),$$

whereas the Gaussian concentration from the preceding part grants

$$\mathbb{P}[\|w\| > r + i] \leq \exp(-(r + i - \sqrt{d})^2/2) \leq \exp(-(r - \sqrt{d})^2/2) \exp(-i^2/2),$$

whereby

$$
\begin{aligned}
\int_{\|w\|>r} \|w\| \, \mathrm{d}G(w) &\leq (r + 1) \int_{\|w\|>r} \mathrm{d}G(w) + \sum_{i=0}^{\infty} i \int_{\|w\|>r+i} \mathrm{d}G(w) \\
&\leq (r + 1) \exp(-(r - \sqrt{d})^2/2) + \sum_{i=0}^{\infty} i \exp(-(r - \sqrt{d})^2/2) \exp(-i^2/2) \\
&\leq \exp(-(r - \sqrt{d})^2/2) \left[ r + 1 + \sum_{i=0}^{\infty} i \exp(-i^2/2) \right] \\
&\leq \exp(-(r - \sqrt{d})^2/2) \left[ r + 2 \right].
\end{aligned}
$$

The final inequality follows by applying Lemma A.2 with $(a, b, c, x) = (3, \sqrt{d}, 2, r)$

4. Proceeding similarly,

$$\int_{\|w\|>r} \|w\| \, \mathrm{d}G(w) \leq \sum_{i=0}^{\infty} \int_{r+i<\|w\|\leq r+i+1} (r+i+1)^2 \, \mathrm{d}G(w),$$

$$\leq 2(r+1)^2 \int_{\|w\|>r} \mathrm{d}G(w) + 2\sum_{i=0}^{\infty} i^2 \int_{\|w\|>r+i} \mathrm{d}G(w)$$

$$\leq 2(r+1)^2 \exp(-(r-\sqrt{d})^2/2) + 2\sum_{i=0}^{\infty} i^2 \exp(-(r-\sqrt{d})^2/2)\exp(-i^2/2)$$

$$\leq 2\exp(-(r-\sqrt{d})^2/2)\left[(r+1)^2 + \sum_{i=0}^{\infty} i^2 \exp(-i^2/2)\right]$$

$$\leq 2\exp(-(r-\sqrt{d})^2/2)\left[r+2\right]^2.$$

The final inequality follows by applying Lemma A.2 with $(a,b,c,x) = (7,\sqrt{d},2,r)$.

$\square$

**Fourier transforms.** The convention for the Fourier transform used here is

$$\hat{f}(w) = \int f(x)\exp(2\pi i w^{\mathsf{T}} x)\,\mathrm{d}x;$$

see for instance (Folland, 1999, Section 8.8) for a discussion of other conventions, and the resulting tradeoffs. Note also the polar decomposition notation $\hat{f}(w) = |\hat{f}(w)|\exp(2\pi i \theta_f(w))$ with $|\theta_f(w)| \leq 1$. The following lemma collects a few properties used throughout.

**Lemma A.3.**    *1.* $|\hat{f}| \leq \|f\|_{L_1}$.

2. $\widehat{f * g} = \hat{f}\hat{g}$ *and* $|\widehat{f * g}| \leq \|f\|_{L_1}|\hat{g}|$.

3. *Let* $\alpha > 0$ *be given and define* $\phi := (2\pi\alpha)^{-1}$. *Then* $|\hat{G}_\alpha| = \hat{G}_\alpha$ *(meaning* $\hat{G}_\alpha$ *has no radial component, thus* $\theta_{G_\alpha}(w) = 0$*), and*

$$\hat{G}_\alpha(w) = (2\pi\alpha^2)^{-d/2}G_\phi(w) = (2\pi\phi^2)^{d/2}G_\phi(w) = (2\pi)^{d/2}G(w/\phi).$$

*Proof.*    1. Directly,

$$|\hat{h}(w)| \leq \int |h(x)| \cdot |\exp(2\pi i w^{\mathsf{T}} x)|\,\mathrm{d}x \leq \|h\|_{L_1},$$

2. The first equality is standard (Folland, 1999, Theorem 8.22c), and the inequality combines it with the preceding bound.

3. The form of $\hat{G}_\alpha$ and the first displayed inequality are standard (Folland, 1999, Proposition 8.24). The second and third inequalities use the choice of $\phi$ and the form of $G_\phi$.

$\square$

**ReLU representation.** Lastly, the exact ReLU representation constructions (e.g., Lemma 3.1) will use the following folklore lemma to write a univariate twice continuously differentiable function as an infinite width ReLU network.

**Lemma A.4.** *Let* $f : \mathbb{R} \to \mathbb{R}$ *be given with* $f(0) = f'(0) = 0$ *and* $f''$ *continuous. For any* $z \geq 0$,

$$f(z) = \int_0^\infty \sigma(z-b)f''(b)\,\mathrm{d}b.$$

*Proof.* Using integration by parts,

$$
\begin{aligned}
\int_0^\infty \sigma(z-b)f''(b)\,\mathrm{d}b &= \int_0^z (z-b)f''(b)\,\mathrm{d}b \\
&= z\int_0^z f''(b)\,\mathrm{d}b - \int_0^z bf''(b)\,\mathrm{d}b \\
&= zf'(b)|_0^z - \left(bf'(b)|_0^z - \int_0^z f'(b)\,\mathrm{d}b\right) \\
&= zf'(z) - zf'(0) - \big(zf'(z) - 0 - f(z) + f(0)\big) \\
&= f(z).
\end{aligned}
$$

$\square$

## B    SAMPLING TOOLS: MAUREY'S LEMMA AND CO-VC DIMENSION

This section collects various sampling tools used as a basis for Section 2. First is a proof of Lemma 2.5, which here is combined with an application of McDiarmid's inequality to give a high probability guarantee.

*Proof of Lemma 2.5.* Following the usual Maurey scheme (Pisier, 1980),

$$
\begin{aligned}
\mathop{\mathbb{E}}_{(v_j)_{j=1}^m}\left\| f - \frac{1}{m}\sum_j g_j \right\|_{L_2(P)}^2 &= \frac{1}{m^2}\mathop{\mathbb{E}}_{(v_j)_{j=1}^m}\left\| \sum_j (f-g_j) \right\|_{L_2(P)}^2 \\
&= \frac{1}{m^2}\mathop{\mathbb{E}}_{(v_j)_{j=1}^m}\sum_j \| f - g_j \|_{L_2(P)}^2 \\
&= \frac{1}{m}\mathop{\mathbb{E}}_{v_1}\| f - g_1 \|_{L_2(P)}^2 \\
&= \frac{1}{m}\mathop{\mathbb{E}}_{v_1}\left( \| g_1 \|_{L_2(P)}^2 - \| f \|_{L_2(P)}^2 \right) \\
&\leq \frac{1}{m}\mathbb{E}_{v_1}\| g_1 \|_{L_2(P)}^2 \\
&\leq \frac{1}{m}\sup_{v\in\mathcal{V}}\| g(\cdot;v) \|_{L_2(P)}^2 .
\end{aligned}
$$

The high probability bound will follow from McDiarmid's inequality. To establish the bounded differences property, define

$$
F(V) := F((v_1,\dots,v_m)) = \left\| f - \frac{1}{m}\sum_j g(\cdot;v_j) \right\|_{L_2(P)},
$$

and note from the general metric space inequality $\big|\|p\| - \|q\|\big| \leq \|p-q\|$ that for any $V = (v_1,\dots,v_m)$ and $V' = (v_1',\dots,v_m')$ differing only on a single $v_k \neq v_k'$,

$$
\begin{aligned}
\big\| F(V) - F(V') \big\| &\leq \left\| \frac{1}{m}\sum_j g(\cdot;v_j) - \frac{1}{m}\sum_j g(\cdot;v_j') \right\|_{L_2(P)} \\
&= \frac{1}{m}\big\| g(\cdot;v_k) - g(\cdot;v_k') \big\|_{L_2(P)} \\
&\leq \frac{1}{m}\left( \big\| g(\cdot;v_k) \big\|_{L_2(P)} + \big\| g(\cdot;v_k') \big\|_{L_2(P)} \right) \\
&\leq \frac{2}{m}\sup_{v\in\mathcal{V}}\big\| g(\cdot;v) \big\|_{L_2(P)} .
\end{aligned}
$$

Thus, with probability at least $1 - \eta$, McDiarmid's inequality grants,

$$F(V) \le \mathbb{E}_V F(V) + \sup_{v \in \mathcal{V}} \|g(\cdot; v)\|_{L_2(P)} \sqrt{\frac{2 \ln(1/\eta)}{m}},$$

and the statement follows by Jensen's inequality, specifically $\mathbb{E}_V F(V) \le \sqrt{\mathbb{E}_V F(V)^2}$. $\qquad \square$

Maurey's lemma also applies to sampling from signed densities in the sense of Definition 4.4.

**Lemma B.1.** *Let $f(x) = \int p(\tilde{w}) g(\langle \tilde{w}, \tilde{x} \rangle) \, \mathrm{d}\tilde{w}$ be given with $\|p\|_{L_1} < \infty$ and $p$ is supported on a ball of radius $B$, and let $((s_j, \tilde{w}_j))_{j=1}^m$ be sampled from $p$ as in Definition 4.4, and define $g_j(x) := g(\langle \tilde{w}, \tilde{x} \rangle)$. With probability at least $1 - \eta$,*

$$\left\| f - \frac{\|p\|_{L_1}}{m} \sum_{j=1}^m s_j g_j \right\|_{L^2(P)} \le \sup_{\|\tilde{w}\| \le B} \|g(\langle \tilde{w}, \cdot \rangle)\|_{L_2(P)} \|p\|_{L_1} \left[ \frac{1 + \sqrt{2 \ln(1/\eta)}}{\sqrt{m}} \right].$$

*Proof.* Since

$$\int p(\tilde{w}) g(\langle \tilde{w}, \tilde{x} \rangle) \, \mathrm{d}\tilde{w} = \|p\|_{L_1} \int \mathrm{sgn}(p) \frac{|p(\tilde{w})|}{\|p\|_{L_1}} g(\langle \tilde{w}, \tilde{x} \rangle) \, \mathrm{d}\tilde{w} = \|p\|_{L_1} \mathop{\mathbb{E}}_{s, \tilde{w}} s g(\langle \tilde{w}, \tilde{x} \rangle),$$

the sampling procedure indeed provides an unbiased estimate of the integral, and thus by Maurey's Lemma (cf. Lemma 2.5), with probability at least $1 - \eta$,

$$\left\| f - \frac{\|p\|_{L_1}}{m} \sum_{j=1}^m s_j g_j \right\|_{L^2(P)} = \|p\|_{L_1} \left\| \mathbb{E}_{s, \tilde{w}} s g(\langle \tilde{w}, \cdot \rangle) - \frac{1}{m} \sum_{j=1}^m s_j g_j \right\|_{L^2(P)}$$

$$\le \|p\|_{L_1} \sup_{\substack{s \in \{\pm 1\} \\ \|\tilde{w}\| \le B}} \|s g(\langle \tilde{w}, \cdot \rangle)\|_{L_2(P)} \left[ \frac{1 + \sqrt{2 \ln(1/\eta)}}{\sqrt{m}} \right]$$

$$= \|p\|_{L_1} \sup_{\|\tilde{w}\| \le B} \|g(\langle \tilde{w}, \cdot \rangle)\|_{L_2(P)} \left[ \frac{1 + \sqrt{2 \ln(1/\eta)}}{\sqrt{m}} \right].$$

$\qquad \square$

Lastly, here is a uniform norm analog of the preceding $L_2(P)$ signed density sampling bound. Interestingly, the bound only gives a $\sqrt{d}$ degradation with $\sigma'$, and no degradation for $\sigma$. The method of proof is to use uniform convergence, but with data and parameters switched; consequently, this has been called "co-VC dimension" (Gurvits & Koiran, 1995; Sun et al., 2018). The proof is somewhat more complicated than the proof of the Maurey lemma, and in particular needs to be a bit more attentive to the fine-grained structure of the functions being sampled.

**Lemma B.2.** *Let density $p : \mathbb{R}^{d+1} \to \mathbb{R}$ with $\|p\|_{L_1} < \infty$, and let $((s_j, w_j))_{j=1}^m$ be a sample from $p$ in the sense of Definition 4.4.*

    *1. With probability at least $1 - 2\eta$,*

$$\sup_{\|x\| \le 1} \left| \int p(\tilde{w}) \sigma'(\langle \tilde{w}, \tilde{x} \rangle) \, \mathrm{d}\tilde{w} - \frac{\|p\|_{L_1}}{m} \sum_j s_j \sigma'(\langle \tilde{w}_j, \tilde{x} \rangle) \right| \le \frac{\|p\|_{L_1}}{\sqrt{m}} \left[ \sqrt{8(d+1) \ln(m+1)} + \sqrt{\ln(1/\eta)} \right].$$

    *2. Suppose $p$ is supported on the set $\mathcal{W} := \{ \tilde{w} \in \mathbb{R}^{d+1} : \|w\| \le r, |b| \le \|w\| \}$. With probability at least $1 - 2\eta$,*

$$\sup_{\|x\| \le 1} \left| \int p(\tilde{w}) \sigma(\langle \tilde{w}, \tilde{x} \rangle) \, \mathrm{d}\tilde{w} - \frac{\|p\|_{L_1}}{m} \sum_j s_j \sigma(\langle \tilde{w}_j, \tilde{x} \rangle) \right| \le \frac{4r \|p\|_{L_1}}{\sqrt{m}} \left[ 1 + \sqrt{\ln(1/\eta)} \right].$$

*Proof of Lemma B.2.* In both cases, letting $g$ denote either of $\sigma'$ or $\sigma$,

$$\sup_{\|x\|\leq 1}\left|\int p(\tilde{w})g(\langle\tilde{w},\tilde{x}\rangle)\,\mathrm{d}\tilde{w}-\frac{\|p\|_{L_1}}{m}\sum_j s_j g(\langle\tilde{w}_j,\tilde{x}\rangle)\right|$$

$$=\|p\|_{L_1}\sup_{\|x\|\leq 1}\left|\mathbb{E}_p sg(\langle\tilde{w},\tilde{x}\rangle)\,\mathrm{d}\tilde{w}-\frac{1}{m}\sum_j s_j g(\langle\tilde{w}_j,\tilde{x}\rangle)\right|,$$

and at this point it is a classical uniform deviations problem, but with the role of parameter and data swapped, an approach which has been used before (sometimes under the heading "co-VC dimension" (Gurvits & Koiran, 1995; Sun et al., 2018)). Continuing, with probability at least $1-2\eta$, standard Rademacher complexity (Shalev-Shwartz & Ben-David, 2014) grants

$$\sup_{\|x\|\leq 1}\left|\mathbb{E}_p sg(\langle\tilde{w},\tilde{x}\rangle)\,\mathrm{d}\tilde{w}-\frac{1}{m}\sum_j s_j g(\langle\tilde{w}_j,\tilde{x}\rangle)\right|\leq 2\mathrm{Rad}\left(\left\{(s_j g(\langle\tilde{w}_j,\tilde{x}\rangle))_{j=1}^m:\|x\|\leq 1\right\}\right)$$

$$+3\sup_{\substack{\tilde{w}\in\mathcal{W}\\\|x\|\leq 1}}|g(\langle\tilde{w},\tilde{x}\rangle)|\sqrt{\frac{\ln(1/\eta)}{2m}}.$$

where $\mathcal{W}$ is a constraint set on $\tilde{w}$ (when $g=\sigma'$, it is $\mathbb{R}^{d+1}$, whereas with $g=\sigma$ it is $|b|\leq\|w\|\leq r$). To simplify further, note that a Rademacher random vector $(\epsilon_1,\ldots,\epsilon_m)$ is distributionally equivalent to $(s_1\epsilon_1,\ldots,s_m\epsilon_m)$ for any fixed vector of signs $(s_1,\ldots,s_m)$, and therefore

$$\mathrm{Rad}\left(\left\{(s_j g(\langle\tilde{w}_j,\tilde{x}\rangle))_{j=1}^m:\|x\|\leq 1\right\}\right)=\frac{1}{n}\mathbb{E}_\epsilon\sup_{\|x\|\leq 1}\sum_j s_j\epsilon_j g(\langle\tilde{w},\tilde{x}\rangle)$$

$$=\frac{1}{n}\mathbb{E}_\epsilon\sup_{\|x\|\leq 1}\sum_j\epsilon_j g(\langle\tilde{w},\tilde{x}\rangle)$$

$$=\mathrm{Rad}\left(\left\{(g(\langle\tilde{w}_j,\tilde{x}\rangle))_{j=1}^m:\|x\|\leq 1\right\}\right).$$

Combining these steps, with probability at least $1-2\eta$,

$$\sup_{\|x\|\leq 1}\left|\int p(\tilde{w})g(\langle\tilde{w},\tilde{x}\rangle)\,\mathrm{d}\tilde{w}-\frac{\|p\|_{L_1}}{m}\sum_j s_j g(\langle\tilde{w}_j,\tilde{x}\rangle)\right|$$

$$\leq\|p\|_{L_1}\left[2\mathrm{Rad}\left(\left\{(g(\langle\tilde{w}_j,\tilde{x}\rangle))_{j=1}^m:\|x\|\leq 1\right\}\right)+3\sup_{\substack{\tilde{w}\in\mathcal{W}\\\|x\|\leq 1}}|g(\langle\tilde{w},\tilde{x}\rangle)|\sqrt{\frac{\ln(1/\eta)}{2m}}\right].$$

The proof now splits into two cases $g\in\{\sigma',\sigma\}$, bounding the remaining terms.

1. Since the range of $\sigma'$ is $\{0,1\}$,

$$\sup_{\substack{\tilde{w}\in\mathcal{W}\\\|x\|\leq 1}}|\sigma'(\langle\tilde{w},\tilde{x}\rangle)|\leq 1,$$

and the Rademacher complexity is the VC dimension of linear predictors, thus

$$\mathrm{Rad}\left(\left\{(\sigma'(\langle\tilde{w}_j,\tilde{x}\rangle))_{j=1}^m:\|x\|\leq 1\right\}\right)\leq\sqrt{\frac{2(d+1)\ln(m+1)}{m}}.$$

2. In the case $g=\sigma$, since $\mathcal{W}:=\{\tilde{w}\in\mathbb{R}^{d+1}:\|w\|\leq r,|b|\leq\|w\|\}$,

$$\sup_{\substack{\tilde{w}\in\mathcal{W}\\\|x\|\leq 1}}|\sigma(\langle\tilde{w},\tilde{x}\rangle)|\leq\sup_{\substack{\|w\|\leq r\\|b|\leq\|w\|\\\|x\|\leq 1}}|w^\intercal x+b|\leq 2r.$$

Moreover, the Rademacher complexity is a standard combination of the Lipschitz composition rule and linear prediction rules (Shalev-Shwartz & Ben-David, 2014), and thus

$$\mathrm{Rad}\left(\left\{(\sigma'(\langle\tilde{w}_j,\tilde{x}\rangle))_{j=1}^m : \|x\| \leq 1\right\}\right) \leq \mathrm{Rad}\left(\left\{(\langle\tilde{w}_j,\tilde{x}\rangle)_{j=1}^m : \|x\| \leq 1\right\}\right) \leq \frac{4r}{\sqrt{m}}.$$

$\square$

## C    DEFERRED PROOFS FROM SECTION 2

For convenience throughout this appendix, define

$$B := \sup_{\tilde{w}} \|\mathcal{T}(\tilde{w})\| \qquad \text{and} \qquad B_\epsilon := \sup_{\tilde{w},s} \|\mathcal{T}_\epsilon(\tilde{w},s)\|_2 \leq \frac{B}{\epsilon\sqrt{m}} + R.$$

The first step is to prove eq. (2.2), restated here as follows.

**Lemma C.1.** *With probability at least $1 - \eta$ over $(\widetilde{W}, S)$,*

$$\left\| \sum_{j=1}^m \langle\tau_j, \phi_j(\cdot)\rangle - \mathbb{E}_{\tilde{w}} \langle\mathcal{T}(\tilde{w}), \Phi(\cdot;\tilde{w})\rangle \right\|_{L^2(P)} \leq \epsilon B_\epsilon \left[\sqrt{2} + 2\sqrt{\ln(1/\eta)}\right].$$

*Proof.* The proof proceeds by applying Maurey sampling (cf. Lemma 2.5) to the functions $g_j(x) := m\langle\tau_j, \phi_j(x)\rangle$, noting by Lemma 2.4 that

$$f(x) := \mathbb{E}_{\widetilde{W},S} \frac{1}{m} \sum_j g_j(x) = \mathbb{E}_{\widetilde{W},S} \sum_j \langle\tau_j, \phi_j(x)\rangle = \mathbb{E}_{\widetilde{W},S} \left\langle\mathcal{T}_\epsilon(\widetilde{W}, S), \Phi_\epsilon(x;\widetilde{W}, S)\right\rangle = \mathbb{E}_{\tilde{w}} \langle\mathcal{T}(\tilde{w}), \Phi(x;\tilde{w})\rangle.$$

Applying Lemma 2.5, with probability at least $1 - \eta$,

$$\left\| f - \frac{1}{m} \sum_{j=1}^m g_j \right\|_{L^2(P)} \leq \frac{\sup_{\tilde{w},s} m\|\langle\mathcal{T}_\epsilon(\tilde{w},s), \Phi_\epsilon(\cdot;\tilde{w},s)\rangle\|_{L_2(P)}}{\sqrt{m}} \left[1 + \sqrt{2\ln(1/\eta)}\right],$$

where

$$\sup_{\tilde{w},s} \|\langle\mathcal{T}_\epsilon(\tilde{w},s), \Phi_\epsilon(\cdot;\tilde{w},s)\rangle\|_{L_2(P)}^2 \leq \sup_{\tilde{w},s} \mathbb{E}_x \|\mathcal{T}_\epsilon(\tilde{w})\|_2^2 \|\Phi_\epsilon(x;\tilde{w},s)\|_2^2 \leq \frac{2\epsilon^2 B_\epsilon^2}{m}.$$

$\square$

Next, the restatement of eq. (2.3) is as follows.

**Lemma C.2.** *With probability at least $1 - \eta$, If $R \geq \sqrt{d} + 2\sqrt{\ln\left(\frac{\epsilon\sqrt{m\pi}}{B\sqrt{2}}\right)}$, then with probability at least $1 - \eta$,*

$$\left\| \sum_j \langle\tau_j, \phi_j(\cdot)\rangle - \sum_j \frac{s_j\epsilon}{\sqrt{m}}\sigma(\langle\tau_j, \tilde{x}\rangle) \right\|_{L_2(P)} \leq \frac{B_\epsilon B}{m\sqrt{\pi}} + \epsilon B_\epsilon \left[\sqrt{2} + 2\sqrt{\ln(1/\eta)}\right].$$

Recall that the proof of Lemma C.2, as discussed in the body, must calculate the fraction of activations which change, which was collected into Lemma 2.6.

*Proof of Lemma 2.6.* Consider an idealized $\mathcal{T}_\epsilon'$ which does not truncate, whereby

$$\left| |\langle\mathcal{T}_\epsilon'(\tilde{w}), \tilde{x}\rangle| - |\langle\tilde{w}, \tilde{x}\rangle| \right| \leq \left| \frac{\langle\mathcal{T}(\tilde{w}), \tilde{x}\rangle}{\epsilon\sqrt{m}} + \langle\tilde{w}, \tilde{x}\rangle - \langle\tilde{w}, \tilde{x}\rangle \right| \leq \frac{B\|\tilde{x}\|}{\epsilon\sqrt{m}}.$$

The event $\left[\mathrm{sgn}(\langle \tilde{w}, \tilde{x}\rangle) \neq \mathrm{sgn}(\langle \mathcal{T}'_\epsilon(\tilde{w}), \tilde{x}\rangle)\right]$ implies the event $\left[|\langle \tilde{w}, \tilde{x}\rangle| \leq {}^{B\|\tilde{x}\|}/_{\epsilon\sqrt{m}}\right]$, and thus, additionally using rotational invariance of the Gaussian,

$$
\begin{aligned}
\mathbb{E}_{\tilde{w}} \left| \sigma'(\langle \tilde{w}, \tilde{x}\rangle) - \sigma'(\langle \mathcal{T}'_\epsilon(\tilde{w}), \tilde{x}\rangle) \right| &= \mathbb{P}_{\tilde{w}} \left[ \mathrm{sgn}(\langle \tilde{w}, \tilde{x}\rangle) \neq \mathrm{sgn}(\langle \mathcal{T}'_\epsilon(\tilde{w}), \tilde{x}\rangle) \right] \\
&\leq \mathbb{P}_{\tilde{w}} \left[ |\langle \tilde{w}, \tilde{x}\rangle| \leq {}^{B\|\tilde{x}\|}/_{\epsilon\sqrt{m}} \right] \\
&= \mathbb{P}_{\tilde{w}} \left[ |w_1| \cdot \|\tilde{x}\| \leq {}^{B\|\tilde{x}\|}/_{\epsilon\sqrt{m}} \right] \\
&= \frac{1}{\sqrt{2\pi}} \int_{-B/(\epsilon\sqrt{m})}^{B/(\epsilon\sqrt{m})} e^{-z^2/2} \, \mathrm{d}z \\
&\leq \frac{B}{\epsilon} \sqrt{\frac{2}{m\pi}}.
\end{aligned}
$$

Returning to the general case with truncation, by Lemma A.1, using the assumed lower bound on $R$,

$$
\mathbb{P}[\|\tilde{w}\| > R] \leq \exp(-(R - \sqrt{d})^2/2) \leq \frac{B}{\epsilon} \sqrt{\frac{2}{m\pi}},
$$

which gives the final bound via triangle inequality. $\qquad \square$

With Lemma 2.6 in hand, the proof of Lemma C.2 is now an application of Maurey's lemma, with an invocation of positive homogeneity to massage terms.

*Proof of Lemma C.2.* The approach is once again to apply Maurey sampling (cf Lemma 2.5). To this end, define

$$
g(x; \tilde{w}, s) := m\left( \langle \mathcal{T}_\epsilon(\tilde{w}, s), \Phi_\epsilon(x; \tilde{w}, s)\rangle - \frac{s\epsilon}{\sqrt{m}} \sigma\left( \langle \mathcal{T}_\epsilon(\tilde{w}, s), \tilde{x}\rangle \right) \right) \quad \text{and} \quad f(x) = \mathbb{E}_{\tilde{w}, s}\, g(x; \tilde{w}, s),
$$

as well as $g_j(x) := g(x; \tilde{w}_j, s_j)$. Using this notation, the goal of this proof is to upper bound

$$
\left\| \sum_j \langle \tau_j, \phi_j(\cdot)\rangle - \sum_j \frac{s_j \epsilon}{\sqrt{m}} \sigma(\langle \tau_j, \cdot\rangle) \right\|_{L_2(P)} = \left\| \frac{1}{m} \sum_j g_j \right\|_{L_2(P)}.
$$

By Lemma 2.5, with probability at least $1 - \eta$,

$$
\begin{aligned}
\left\| \frac{1}{m} \sum_j g_j \right\|_{L_2(P)} &\leq \|f\|_{L_2(P)} + \left\| f - \frac{1}{m} \sum_j g_j \right\|_{L_2(P)} \\
&\leq \|f\|_{L_2(P)} + \sup_{\tilde{w}, s} \|g(\cdot; \tilde{w}, s)\|_{L_2(P)} \left[ \frac{1 + \sqrt{2\ln(1/\eta)}}{\sqrt{m}} \right].
\end{aligned}
$$

To control these terms, fixing any $(\tilde{w}, s)$, it holds by positive homogeneity of $\sigma$ that

$$
\left\| g(x; \tilde{w}, s) \right\|_{L_2(P)}^2 = m^2 \mathbb{E}_x \left\langle \mathcal{T}_\epsilon(\tilde{w}, s), \Phi_\epsilon(x; \tilde{w}, s) - \frac{s\epsilon}{\sqrt{m}} \Phi(x; \mathcal{T}_\epsilon(\tilde{w}, s)) \right\rangle^2 \leq m^2 B_\epsilon^2 \mathbb{E}_x \frac{\epsilon^2 \|x\|^2}{m} \leq 2m\epsilon^2 B_\epsilon^2.
$$

On the other hand, by Lemma 2.6, for any $\|x\| \leq 1$,

$$
\begin{aligned}
|f(x)| &\leq \mathbb{E}_{\tilde{w}, s} \left| \left\langle \mathcal{T}_\epsilon(\tilde{w}, s), \Phi_\epsilon(x; \tilde{w}, s) - \frac{s\epsilon}{\sqrt{m}} \Phi(x; \mathcal{T}_\epsilon(\tilde{w}, s)) \right\rangle \right| \\
&\leq B_\epsilon \mathbb{E}_{\tilde{w}, s} \frac{\epsilon \|\tilde{x}\| \left| \sigma'(\mathcal{T}_\epsilon(\tilde{w})^\mathsf{T}\tilde{x}) - \sigma'(\tilde{w}^\mathsf{T}\tilde{x}) \right|}{\sqrt{m}} \\
&\leq \frac{\epsilon B_\epsilon}{\sqrt{m}} \left( \frac{B}{\epsilon\sqrt{m\pi}} \right),
\end{aligned}
$$

which also upper bounds $\|f\|_{L_2(P)}$. $\qquad \square$

The proof of Theorem 2.1 now follows by combining Lemmas C.1 and C.2.

*Proof of Theorem 2.1.* By Lemma A.1 and a union bound on $(\tilde{w}_1, \ldots, \tilde{w}_m)$, $\max_j \|\tilde{w}_j\| \leq R$, thus

$$\max_j \|\mathcal{T}_\epsilon(\tilde{w}_j) - \tilde{w}_j\| \leq \max_j \frac{\|\mathcal{T}(\tilde{w}_j)\|}{\epsilon\sqrt{m}} \leq \frac{B}{\epsilon\sqrt{m}}.$$

Moreover, $R \geq \sqrt{d} + 2\sqrt{\ln\left(\frac{\epsilon\sqrt{m}\pi}{B\sqrt{2}}\right)}$, and thus the two other bounds are from Lemmas C.1 and C.2.

□

# D   DEFERRED PROOFS FROM SECTION 3

The first core lemma shows how to write a target function $f$ as an infinite-width network via its Fourier transform.

*Proof of Lemma 3.1.* The first steps are the same for $\sigma$ and $\sigma'$, and indeed match the initial steps of (Barron, 1993), namely

$$f(x) - f(0) = \operatorname{Re} \int \exp(2\pi i x^\mathsf{T} w) \hat{f}(w) \, dw$$

$$= \operatorname{Re} \int \exp(2\pi i x^\mathsf{T} w + 2\pi i \theta_f(w)) |\hat{f}(w)| \, dw$$

$$= \int \cos\left(2\pi(x^\mathsf{T} w + \theta_f(w))\right) |\hat{f}(w)| \, dw.$$

For convenience, define $h(z) := \cos(2\pi z)$, whereby

$$f(x) - f(0) = \int h(x^\mathsf{T} w + \theta_f(w)) |\hat{f}(w)| \, dw, \tag{D.1}$$

and the proofs not differ for both activations and from (Barron, 1993).

1. Consider first $\sigma'$. Since $\|x\| \leq 1$, by Cauchy-Schwarz it suffices to approximate $h$ along the interval $[-\|w\| + \theta_f(w), \|w\| + \theta_f(w)]$. By the fundamental theorem of calculus,

$$h(\langle w, x \rangle + \theta_f(w)) - h\left(-\|w\| + \theta_f(w)\right)$$

$$= \int_{-\|w\|+\theta_f(w)}^{\langle w,x\rangle+\theta_f(w)} h'(b) \, db$$

$$= \int h'(b) \mathbb{1}[\langle w, x \rangle + \theta_f(w) \geq b] \mathbb{1}[b \geq -\|w\| + \theta_f(w)] \, db$$

$$= -\int h'(\theta_f(w) - b) \mathbb{1}[x^\mathsf{T} w + b \geq 0] \mathbb{1}[\|w\| \geq b] \, db, \qquad b \mapsto \theta_f(w) - b$$

$$= -\int h'(\theta_f(w) - b) \mathbb{1}[x^\mathsf{T} w + b \geq 0] \mathbb{1}[\|w\| \geq |b|] \, db,$$

where the last step follows since $\mathbb{1}[x^\mathsf{T} w + b \geq 0]$ implies $b \geq -\|w\|$. Plugging this back in to eq. (D.1) and still using $h(z) = \cos(2\pi z)$,

$$f(x) - f(0) = \int |\hat{f}(w)| \cos\left(2\pi(x^\mathsf{T} w + \theta_f(w))\right) \, dw$$

$$= \int |\hat{f}(w)| \left[ h(-\|w\| + \theta_f(w)) - \int h'(\theta_f(w) - b) \mathbb{1}[x^\mathsf{T} w + b \geq 0] \mathbb{1}[\|w\| \geq |b|] \, db \right] \, dw,$$

which after pushing more terms onto the left hand side gives

$$f(x) - f(0) - \int |\hat{f}(w)| h(\theta_f(w) - \|w\|) \, \mathrm{d}w$$

$$= -\iint |\hat{f}(w)| h'(\theta_f(w) - b) \mathbb{1}[x^\mathsf{T}w + b \geq 0] \mathbb{1}[\|w\| \geq |b|] \, \mathrm{d}b \, \mathrm{d}w$$

$$= 2\pi \int |\hat{f}(w)| \sin(2\pi(\theta_f(w) - b)) \sigma'(\langle \tilde{w}, \tilde{x} \rangle) \mathbb{1}[\|w\| \geq |b|] \, \mathrm{d}\tilde{w},$$

which gives $F_\infty = f$ for $\|x\| \leq 1$. To bound the error of $F_r$, note by the form of $F_\infty$ for any $\|x\| \leq 1$ that

$$\left| f(x) - F_r(x) \right| = \left| 2\pi \int_{\|w\| > r} \int |\hat{f}(w)| \sin(2\pi(\theta_f(w) - b)) \sigma'(\langle \tilde{w}, \tilde{x} \rangle) \mathbb{1}[\|w\| \geq |b|] \, \mathrm{d}b \, \mathrm{d}w \right|$$

$$\leq 2\pi \int_{\|w\| > r} \int_{|b| \leq \|w\|} |\hat{f}(w)| |\sin(2\pi(\theta_f(w) - b))| \sigma'(\langle \tilde{w}, \tilde{x} \rangle) \, \mathrm{d}b \, \mathrm{d}w$$

$$\leq 2\pi \int_{\|w\| > r} |\hat{f}(w)| \int_{|b| \leq \|w\|} \mathrm{d}b \, \mathrm{d}w$$

$$= 4\pi \int_{\|w\| > r} \|w\| \cdot |\hat{f}(w)| \, \mathrm{d}w.$$

2. Now consider $\sigma$. Rather than using FTC as above, this proof replaces $h$ with ReLUs via Lemma A.4, which requires a function which is both zero and flat at 0. To this end, define

$$H(b) = h(b + q) - \big( h(q) + bh'(q) \big) \quad \text{with } q = -\|w\| + \theta_f(w),$$

whereby $H(0) = 0 = H'(0)$. Invoking Lemma A.4 on $H$ gives, for any $z := w^\mathsf{T}x + \theta_f(w) \geq q$,

$$h(z) - \big( h(q) + (z - q)h'(q) \big) = H(z - q)$$

$$= \int H''(b) \sigma(z - q - b) \mathbb{1}[b \geq 0] \, \mathrm{d}b$$

$$= \int H''(b) \sigma(w^\mathsf{T}x + \theta_f(w) + \|w\| - \theta_f(w) - b) \mathbb{1}[b \geq 0] \, \mathrm{d}b$$

$$= -\int H''(\|w\| - b) \sigma(w^\mathsf{T}x + b) \mathbb{1}[\|w\| \geq b] \, \mathrm{d}b \qquad b \mapsto \|w\| - b$$

$$= -\int H''(\|w\| - b) \sigma(w^\mathsf{T}x + b) \mathbb{1}[\|w\| \geq |b|] \, \mathrm{d}b,$$

the final equality since $-b > \|w\|$ implies $w^\mathsf{T}x + b \leq \|w\| + b < 0$, thus $\sigma(\tilde{w}^\mathsf{T}\tilde{x}) = 0$ and this case has no effect. Plugging this back into eq. (D.1),

$$f(x) - f(0) = \int |\hat{f}(w)| h(z) \, \mathrm{d}w$$

$$= \int |\hat{f}(w)| \left[ h(q) + (z - q)h'(q) - \int H''(\|w\| - b) \sigma(\tilde{w}^\mathsf{T}\tilde{x}) \mathbb{1}[\|w\| \geq |b|] \, \mathrm{d}b \right] \mathrm{d}w$$

$$= \int |\hat{f}(w)| \left[ h(q) + (w^\mathsf{T}x + \|w\|)h'(q) - \int H''(\|w\| - b) \sigma(\tilde{w}^\mathsf{T}\tilde{x}) \mathbb{1}[\|w\| \geq |b|] \, \mathrm{d}b \right] \mathrm{d}w,$$

which gives $Q_\infty = f$ for $\|x\| \leq 1$ after expanding $h$ and $H$. To bound the error of $Q_r$, for any $\|x\| \leq 1$

$$
\begin{aligned}
\left| f(x) - Q_r(x) \right| &= \left| \int_{\|w\| > r} |\hat{f}(w)| \int H''(\|w\| - b) \sigma(\tilde{w}^\mathsf{T}\tilde{x}) \mathbb{1}[\|w\| \geq |b|] \, \mathrm{d}b \right| \mathrm{d}w \\
&\leq \int_{\|w\| > r} |\hat{f}(w)| \int |H''(\|w\| - b)| \sigma(\tilde{w}^\mathsf{T}\tilde{x}) \mathbb{1}[\|w\| \geq |b|] \, \mathrm{d}b \, \mathrm{d}w \\
&\leq 4\pi^2 \int_{\|w\| > r} |\hat{f}(w)| \int_{-\|w\|}^{\|w\|} \sigma(\tilde{w}^\mathsf{T}\tilde{x}) \, \mathrm{d}b \, \mathrm{d}w \\
&\leq 4\pi^2 \int_{\|w\| > r} |\hat{f}(w)| \int_{-\|w\|}^{\|w\|} (\|w\| + |b|) \, \mathrm{d}b \, \mathrm{d}w \\
&\leq 12\pi^2 \int_{\|w\| > r} \|w\|^2 \cdot |\hat{f}(w)| \, \mathrm{d}w.
\end{aligned}
$$

$\square$

Next, Lemma 3.2 converts Lemma 3.1 into a (random feature) transport map by introducing the fraction $G(\tilde{w})/G(\tilde{w})$.

*Proof of Lemma 3.2.* Starting from the construction in Lemma 3.1, again using $h(z) = \cos(2\pi z)$ for convenience, and manually introducing a factor $G(\tilde{w})$,

$$
\begin{aligned}
f(x) - f(0) &- \int |\hat{f}(w)| h(\theta_f(w) - \|w\|) \, \mathrm{d}w \\
&= -\iint |\hat{f}(w)| h'(\theta_f(w) - b) \mathbb{1}[x^\mathsf{T}w + b \geq 0] \mathbb{1}[\|w\| \geq |b|] \, \mathrm{d}b \, \mathrm{d}w \\
&= -\int \frac{|\hat{f}(w)|}{G(\tilde{w})} h'(\theta_f(w) - b) \sigma'(\langle \tilde{w}, \tilde{x} \rangle) \mathbb{1}[\|w\| \geq |b|] \, \mathrm{d}G(\tilde{w}).
\end{aligned}
$$

To construct $\mathcal{T}_\infty$, rotational invariance of the Gaussian gives $\mathbb{E}\mathbb{1}[\tilde{w}^\mathsf{T}\tilde{x} \geq 0] = 1/2$, thus

$$
f(0) + \int |\hat{f}(w)| h(\theta_f(w) - \|w\|) \, \mathrm{d}w = \int 2 \left[ f(0) + \int |\hat{f}(v)| h(\theta_f(v) - \|v\|) \, \mathrm{d}v \right] \sigma'(\langle \tilde{w}, \tilde{x} \rangle) \, \mathrm{d}G(\tilde{w}),
$$

and transport mapping is $\mathcal{T}_\infty(w, b) = (0, p_\infty(\tilde{w})) \in \mathbb{R}^d \times \mathbb{R}$ with

$$
p_\infty(\tilde{w}) = 2 \left[ f(0) + \int |\hat{f}(v)| h(\theta_f(v) - \|v\|) \, \mathrm{d}v \right] - \frac{|\hat{f}(w)|}{G(\tilde{w})} h'(\theta_f(w) - b) \mathbb{1}[\|w\| \geq |b|],
$$

By construction, $\mathbb{E}_{\tilde{w}} \langle \mathcal{T}_r(\tilde{w}), \Phi(x; \tilde{w}) \rangle = F_r(x)$, and therefore Lemma 3.1 grants for all $\|x\| \leq 1$

$$
f(x) = \mathbb{E}_{\tilde{w}} \langle \mathcal{T}_\infty(\tilde{w}), \Phi(x; \tilde{w}) \rangle \quad \text{and} \quad \left| f(x) - \mathbb{E} \langle \mathcal{T}_r(\tilde{w}), \Phi(x; \tilde{w}) \rangle \right| \leq 4\pi \int_{\|w\| > r} |\hat{f}(w)| \cdot \|w\| \, \mathrm{d}w.
$$

$\square$

With more care (in particular, a crucial change of variable), a much better bound is possible for convolutions with Gaussians.

*Proof of Lemma 3.3.* By Lemma A.3, setting $\phi := (2\pi\sigma)^{-1}$,

$$
|\hat{f}_\alpha(w)| = |\hat{f}(w)| \hat{G}_\alpha(w) = (2\pi)^{d/2} |\hat{f}(w)| G(w/\phi).
$$

Plugging this into Lemma 3.1 and again defining $h(z) := \cos(2\pi z)$ for convenience, but unlike Lemma 3.2 performing a change of variable to directly introduce $G(\tilde{w})$, and then manually introducing

$G(b)$,

$$f_\alpha(x) - f_\alpha(0) - \int |\hat{f}_\alpha(w)| h(\theta_{f_\alpha}(w) - \|w\|) \, dw$$

$$= - \iint |\hat{f}_\alpha(w)| h'(\theta_{f_\alpha}(w) - b) \sigma'(\tilde{w}^\mathsf{T} \tilde{x}) \mathbb{1}[\|w\| \geq |b|] \, db \, dw$$

$$= -(2\pi)^{d/2} \iint |\hat{f}(w)| G(w/\phi) h'(\theta_{f_\alpha}(w) - b) \sigma'(\tilde{w}^\mathsf{T} \tilde{x}) \mathbb{1}[\|w\| \geq |b|] \, db \, dw$$

$$= -(2\pi\phi^2)^{d/2} \phi \int |\hat{f}(\phi w)| G(w) h'(\theta_{f_\alpha}(\phi w) - b) \sigma'(\phi \langle \tilde{w}, \tilde{x} \rangle) \mathbb{1}[\phi \|w\| \geq \phi |b|] \, db \, dw$$

$$= -(2\pi\phi^2)^{(d+1)/2} \int |\hat{f}(\phi w)| e^{b^2/2} h'(\theta_{f_\alpha}(\phi w) - b) \sigma'(\langle \tilde{w}, \tilde{x} \rangle) \mathbb{1}[\|w\| \geq |b|] \, dG(\tilde{w}).$$

As in Lemma 3.2, the transport is constructed by using $\mathbb{E}\sigma'(\langle \tilde{w}, \tilde{x} \rangle) = 1/2$ to model constants: $\mathcal{T}_r(w, b) = (0, \ldots, 0, p_r(\tilde{w}))$, where

$$p_r(\tilde{w}) := 2 \left[ f_\alpha(0) + \int |\hat{f}_\alpha(v)| h(\theta_{f_\alpha}(v) - \|v\|) \, dv \right]$$
$$- (2\pi\phi^2)^{(d+1)/2} |\hat{f}(\phi w)| e^{b^2/2} h'(\theta_{f_\alpha}(\phi w) - b) \mathbb{1} \left[ |b| \leq \|w\| \leq r \right],$$

with $f(x) = \mathbb{E}_{\tilde{w}} \langle \mathcal{T}_\infty(\tilde{w}), \Phi(x; \tilde{w}) \rangle$ for $\|x\| \leq 1$ by construction.

When $r < \infty$, by construction

$$\sup_{\tilde{w}} \|\mathcal{T}_r(\tilde{w})\| \leq 2|f_\alpha(0)| + 2 \int |\hat{f}_\alpha(v)| \, dv + 2\pi(2\pi\phi^2)^{(d+1)/2} \sup_{\substack{\|w\| \leq r \\ |b| \leq \|w\|}} |\hat{f}(\phi w)| e^{b^2/2},$$

where $|\hat{f}(\phi w)| = 1$ when $f_\alpha = G_\alpha$ (meaning $f$ itself is the Dirac at 0), and more generally Lemma A.3 grants $|\hat{f}(\phi w)| \leq \|f(\phi \cdot)\|_{L_1}$; as in the lemma statement, these cases are summarized with $|\hat{f}(\phi w)| \leq M_f$. Plugging this in and simplifying further via Lemma A.3,

$$\sup_{\tilde{w}} \|\mathcal{T}_r(\tilde{w})\| \leq 2 \left| \int f(x) G_\alpha(-x) \, dx \right| + 2(2\pi)^{d/2} \int |\hat{f}(v)| G(v/\phi) \, dv + 2\pi(2\pi\phi^2)^{(d+1)/2} M_f \sup_{|b| \leq r} e^{b^2/2}$$

$$\leq 2 \left[ M + 2(2\pi\phi^2)^{d/2} M_f + 2\pi(2\pi\phi^2)^{(d+1)/2} M_f \sup_{|b| \leq r} e^{b^2/2} \right].$$

For the approximation estimate, for any $\|x\| \leq 1$, the preceding derivation and Lemma A.1 grant

$$\left| f(x) - \mathbb{E} \langle \mathcal{T}_r(\tilde{w}), \Phi(x; \tilde{w}) \rangle \right|$$

$$= \left| \mathbb{E} \langle \mathcal{T}_\infty(\tilde{w}) - \mathcal{T}_r(\tilde{w}), \Phi(x; \tilde{w}) \rangle \right|$$

$$\leq 2\pi(2\pi\phi^2)^{(d+1)/2} \int_{\|w\| > r} \int_{|b| \leq \|w\|} |\hat{f}(\phi w)| |\sin(2\pi(w^\mathsf{T} x + \theta_{f_\alpha}(w))) \sigma'(\tilde{w}^\mathsf{T} \tilde{x})| \, db \, dG(w)$$

$$\leq 2\pi(2\pi\phi^2)^{(d+1)/2} M_f \int_{\|w\| > r} \int_{|b| \leq \|w\|} db \, dG(w)$$

$$\leq 4\pi(2\pi\phi^2)^{(d+1)/2} M_f \int_{\|w\| > r} \|w\| \, dG(w)$$

$$\leq 4\pi(2\pi\phi^2)^{(d+1)/2} M_f (\sqrt{d} + 3) \exp\left( -(r - \sqrt{d})^2/4 \right).$$

$\square$

# E   DEFERRED PROOFS FROM SECTION 4

The first proof is of the approximation properties of Gaussian convolution; as stated in the body, the proof proceeds by splitting the error into two terms, one for nearby points, the other for distant points.

*Proof of Lemma 4.2.* Splitting the integral into two terms, for any $\|x\| \leq 1$,

$$
\begin{aligned}
\left| f(x) - (f_{|\delta} * G_\alpha)(x) \right| &= \left| \int f_{|\delta}(x) G_\alpha(z)\, \mathrm{d}z - \int f_{|\delta}(z) G_\alpha(x-z)\, \mathrm{d}z \right| \\
&= \left| \int f_{|\delta}(x) G_\alpha(z)\, \mathrm{d}z - \int f_{|\delta}(x-z) G_\alpha(z)\, \mathrm{d}z \right| \\
&\leq \int \left| f_{|\delta}(x) - f_{|\delta}(x-z) \right| G_\alpha(z)\, \mathrm{d}z \\
&= \int_{\|z\| \leq \delta} \left| f_{|\delta}(x) - f_{|\delta}(x-z) \right| G_\alpha(z)\, \mathrm{d}z \\
&\quad + \int_{\|z\| > \delta} \left| f_{|\delta}(x) - f_{|\delta}(x-z) \right| G_\alpha(z)\, \mathrm{d}z.
\end{aligned}
$$

Analyzing these terms separately, the definition of $\omega_f(\delta)$ gives

$$
\int_{\|z\| \leq \delta} \left| f_{|\delta}(x) - f_{|\delta}(x-z) \right| G_\alpha(z)\, \mathrm{d}z \leq \int_{\|z\| \leq \delta} \omega_f(\delta) G_\alpha(z)\, \mathrm{d}z \leq \omega_f(\delta),
$$

whereas Gaussian concentration (cf. Lemma A.1) gives

$$
\int_{\|z\| > \delta} \left| f_{|\delta}(x) - f_{|\delta}(x-z) \right| G_\alpha(z)\, \mathrm{d}z \leq 2M\, \mathbb{P}[\|\alpha z\| > \delta] \leq 2M \exp(-(\delta/\alpha - \sqrt{d})^2/2) \leq \omega_f(\delta).
$$

$\square$

This now combines with Lemma 3.3 to prove Theorem 4.3.

*Proof of Theorem 4.3.* Plugging the choice of $r$ into Lemma 3.3, for any $\|x\| \leq 1$,

$$
\left| f(x) - \mathbb{E} \left\langle \mathcal{T}_r(\tilde{w}), \Phi(x; \tilde{w}) \right\rangle \right| \leq 4\pi (2\pi\phi^2)^{(d+1)/2} M_f(\sqrt{d} + 3) \exp\left( -(r - \sqrt{d})^2/4 \right) \leq \omega_f(\delta).
$$

Moreover, plugging $r$ into the estimate on $\sup_{\tilde{w}} \|\mathcal{T}_r(\tilde{w})\|$ provided by Lemma 3.3 gives

$$
\sup_{\tilde{w}} \|\mathcal{T}_r(\tilde{w})\| \leq 2 \left[ M + (2\pi\phi^2)^{d/2} M_f \left( 1 + \sqrt{2\pi^3 \phi^2} e^{r^2/2} \right) \right],
$$

where Lemma A.3 and the choice of $r$ give

$$
M_f \leq \|f_{|\delta}\|_{L_1}, \qquad \text{and} \qquad e^{r^2/2} \leq e^d e^{(r - \sqrt{d})^2},
$$

where

$$
e^{(r - \sqrt{d})^2} = \left( \frac{4\pi (2\pi\phi^2)^{(d+1)/2} M_f(\sqrt{d} + 3)}{\omega_f(\delta)} \right)^4 = \mathcal{O}\left( \left( \frac{M_f \sqrt{d}}{\omega_f(\delta) \alpha^{d+1}} \right)^4 \right),
$$

and noting moreover that $\alpha = \widetilde{\mathcal{O}}(\delta/\sqrt{d})$.

$\square$

To close this section comes the full version of Theorem 4.5, which gives explicit constructions for both threshold $\sigma'$ and ReLU $\sigma$. Interestingly, in the case of $\sigma'$, it is not necessary to truncate the density, as is the case everywhere else in this work.

**Theorem E.1.** *As in Lemma 4.2, let $f : \mathbb{R}^d \to \mathbb{R}$ and $\delta > 0$ be given, and define*

$$
M := \sup_{\|x\| \leq 1 + \delta} |f(x)|, \qquad f_{|\delta}(x) := f(x) \mathbb{1}[\|x\| \leq 1 + \delta], \qquad \alpha := \frac{\delta}{\sqrt{d} + \sqrt{2 \ln(2M/\omega_f(\delta))}}.
$$

*Let $G_\alpha$ denote a Gaussian with the preceding variance $\alpha^2$, and define $h := f_{|\delta} * G_\alpha$ with Fourier transform $\hat{h}$ satisfying radial decomposition $\hat{h}(w) = |\hat{h}(w)| \exp(2\pi i \theta_h(w))$. Lastly, let $P$ be a probability measure supported on $\|x\| \leq 1$.*

1. *Additionally define*

$$c_1 := h(0) + \int |\hat{h}(w)| \cos\left(2\pi(\theta_h(w) - \|w\|)\right) dw, \qquad p_1 := 2\pi|\hat{h}(w)| \sin(2\pi(\theta_h(w) - b))\mathbb{1}\left[|b| \leq \|w\|\right].$$

*Then*

$$|c_1| \leq M + \|f_{|\delta}\|_{L_1}(2\pi\alpha^2)^{d/2}, \qquad \text{and} \qquad \|p_1\|_{L_1} \leq 2\|f_{|\delta}\|_{L_1}\sqrt{\frac{2\pi d}{(2\pi\alpha^2)^{d+1}}},$$

*and with probability at least $1 - 3\eta$ over a draw of $((s_j, \tilde{w}_j))_{j=1}^m$ from $p_1$ (cf. Definition 4.4),*

$$\left\| f - \left[ c_1 + \|p_1\|_{L_1} \sum_{j=1}^m s_j \sigma'(\langle \tilde{w}_j, \cdot \rangle) \right] \right\|_{L_2(P)} \leq 2\omega_f(\delta) + \|p_1\|_{L_1} \left[ \frac{1 + \sqrt{2\ln(1/\eta)}}{\sqrt{m}} \right],$$

$$\sup_{\|x\| \leq 1} \left| f(x) - \left[ c_1 + \|p_1\|_{L_1} \sum_{j=1}^m s_j \sigma'(\langle \tilde{w}_j, x \rangle) \right] \right| \leq 2\omega_f(\delta) + \frac{\|p\|_{L_1}}{\sqrt{m}} \left[ \sqrt{8(d+1)\ln(m+1)} + \sqrt{\ln(1/\eta)} \right].$$

2. *Additionally define*

$$c_2 := f(0)f(0) + \int |\hat{h}(w)| \left[ \cos(2\pi(\theta_h(w) - \|w\|)) - 2\pi\|w\| \sin(2\pi(\theta_h(w) - \|w\|)) \right] dw,$$

$$a_2 := \int w|\hat{h}(w)| \, dw,$$

$$r_2 := \sqrt{d} + 2\sqrt{\ln \frac{24\pi^2(\sqrt{d} + 7)^2\|f_{|\delta}\|_{L_1}}{\omega_f(\delta)}},$$

$$p_2(\tilde{w}) := 4\pi^2|\hat{h}(w)| \cos(2\pi(\|w\| - b))\mathbb{1}[|b| \leq \|w\| \leq r_2],$$

*and for convenience create fake (weight, bias, sign) triples*

$$(w, b, s)_{m+1} := (0, |c_2|, m \cdot \mathrm{sgn}(c_2)), \quad (w, b, s)_{m+2} := (a_2, 0, +m), \quad (w, b, s)_{m+3} := (-a_2, 0, -m).$$

*Then*

$$\|a_2\|_2 \leq \sqrt{d}\|f_{|\delta}\|_{L_1}\phi(2\pi\alpha^2)^{-d/2},$$

$$\|p_2\|_{L_1} \leq 2\|f_{|\delta}\|_{L_1}\sqrt{\frac{(2\pi)^3 d}{(2\pi\alpha^2)^{d+1}}},$$

$$|c_2| \leq M + 2\sqrt{d}\|f_{|\delta}\|_{L_1}(2\pi\alpha^2)^{-d/2},$$

*and with probability at least $1 - 3\eta$ over a draw of $((s_j, \tilde{w}_j))_{j=1}^m$ from $p_2$ (cf. Definition 4.4),*

$$\left\| f - \frac{1}{m} \sum_{j=1}^{m+3} s_j \sigma(\langle \tilde{w}_j, \cdot \rangle) \right\|_{L_2(P)} \leq 3\omega_f(\delta) + r_2\|p\|_{L_1} \left[ \frac{1 + \sqrt{2\ln(1/\eta)}}{\sqrt{m}} \right],$$

$$\sup_{\|x\| \leq 1} \left| f(x) - \frac{1}{m} \sum_{j=1}^{m+3} s_j \sigma(\langle \tilde{w}_j, \cdot \rangle) \right| \leq 3\omega_f(\delta) + \frac{4r_2\|p\|_{L_1}}{\sqrt{m}} \left[ 1 + \sqrt{\ln(1/\eta)} \right].$$

*Proof.* 1. By Lemma 3.1 and the choice of $b_1$, for any $\|x\| \leq 1$,

$$h(x) = c_1 + \int p_1(\tilde{w})\sigma'(\langle \tilde{w}, x \rangle) \, d\tilde{w},$$

thus by Lemma 4.2 and Lemma B.1, defining $h_j := \|p\|_{L_1}\sigma'(\langle \tilde{w}_j, \cdot\rangle)$ for convenience, with probability at least $1 - \eta$,

$$\left\| f - (c_1 + \sum_j h_j/m) \right\|_{L_2(P)} \leq \|f - h\|_{L_2(P)} + \left\| h - (c_1 + \sum_j h_j/m) \right\|_{L_2(P)}$$

$$\leq 2\omega_f(\delta) + \|p_1\|_{L_1} \sup_{\|\tilde{w}\| \leq r_2} \|\sigma'(\langle \tilde{w}, \cdot\rangle\|_{L_2(P)} \left[ \frac{1 + \sqrt{2\ln(1/\eta)}}{\sqrt{m}} \right]$$

$$\leq 2\omega_f(\delta) + \|p_1\|_{L_1} \left[ \frac{1 + \sqrt{2\ln(1/\eta)}}{\sqrt{m}} \right].$$

Similarly, the uniform norm bound follows by Lemma 4.2 and Lemma B.2: with probability at least $1 - 2\eta$, for any $\|x\| \leq 1$,

$$\left| f(x) - (c_1 + \sum_j h_j(x)/m) \right| \leq \left| f(x) - h(x) \right| + \left| h(x) - (c_1 + \sum_j h_j(x)/m) \right|$$

$$\leq 2\omega_f(\delta) + \frac{\|p_1\|_{L_1}}{\sqrt{m}} \left[ \sqrt{8(d+1)\ln(m+1)} + \sqrt{\ln(1/\eta)} \right].$$

For the estimates on $|c_1|$ and $\|p_1\|_{L_1}$, note setting $\phi := (2\pi\alpha)^{-1}$, note by Lemma A.3 and a change of variable $w \mapsto \phi w$ and Lemma A.1 that

$$\|p_1\|_{L_1} \leq 2\pi \int |\widehat{f_{|\delta} * G_\alpha}(w)| \int \mathbb{1}[|b| \leq \|w\|] \, db \, dw$$

$$\leq 4\pi \|f_{|\delta}\|_{L_1} \int \|w\| (2\pi\phi^2)^{d/2} G_\phi(w) \, dw$$

$$= 4\pi (2\pi)^{d/2} \|f_{|\delta}\|_{L_1} \int \|\phi w\| \phi^d G(w) \, dw$$

$$\leq 4\pi (2\pi)^{d/2} \phi^{d+1} \|f_{|\delta}\|_{L_1} \int \|w\| G(w) \, dw$$

$$\leq 4\sqrt{d}\pi (2\pi)^{d/2} \phi^{d+1} \|f_{|\delta}\|_{L_1},$$

$$\leq 2\|f_{|\delta}\|_{L_1} \sqrt{\frac{2\pi d}{(2\pi\alpha^2)^{d+1}}}.$$

Similarly,

$$|c_1| \leq M + \|f_{|\delta}\|_{L_1} \int \hat{G}_\alpha(w) \, dw \leq M + \|f_{|\delta}\|_{L_1} (2\pi\phi^2)^{d/2}.$$

2. By Lemma 3.1 and Lemma A.1 and the various chosen parameters, for any $\|x\| \leq 1$,

$$\left| b_2 + \langle x, a_2\rangle + \int p_2(\tilde{w})\sigma(\langle \tilde{w}, x\rangle) \, d\tilde{w} - h(x) \right| \leq 12\pi^2 \int_{\|w\| > r_2} \|w\|^2 |\hat{h}(w)| \, dw$$

$$\leq 24\pi^2 (\sqrt{d} + 7)^2 \exp(-(r_2 - \sqrt{d})^2/4)$$

$$\leq \omega_f(\delta).$$

Thus by Lemma 4.2 and Lemma B.1, defining $h_j := \|p\|_{L_1} s_j \sigma(\langle \tilde{w}_j, \cdot\rangle)$ for convenience, with probability at least $1 - \eta$,

$$\left\| f - \sum_{j=1}^{m+3} h_j/m \right\|_{L_2(P)} \leq \|f - h\|_{L_2(P)} + \left\| f - (b_2 + (\cdot)^\top c_2 + \mathbb{E}_{p_2} s_1 h_1\| + \left\| \mathbb{E}_{p_2} s_1 h_1 - \sum_{j=1}^m s_j h_j/m \right\|_{L_2(P)}$$

$$\leq 3\omega_f(\delta) + \|p_2\|_{L_1} \sup_{\|\tilde{w}\| \leq r_2} \|\sigma(\langle \tilde{w}, \cdot\rangle\|_{L_2(P)} \left[ \frac{1 + \sqrt{2\ln(1/\eta)}}{\sqrt{m}} \right]$$

$$\leq 3\omega_f(\delta) + 2r_2 \|p\|_{L_1} \left[ \frac{1 + \sqrt{\ln(1/\eta)}}{\sqrt{m}} \right].$$

Similarly, the uniform norm bound follows by Lemma 4.2 and Lemma B.2: with probability at least $1 - 2\eta$, for any $\|x\| \leq 1$,

$$\left| f(x) - \sum_{j=1}^{m+3} h_j(x)/m \right| \leq \left| f(x) - h(x) \right| + \left| f(x) - (b_2 + x^\intercal c_2 + \mathbb{E}_{p_2} s_1 h_1(x) \right| + \left| \mathbb{E}_{p_2} s_1 h_1(x) - \sum_{j=1}^{m} s_j h_j(x)/m \right|$$

$$\leq 3\omega_f(\delta) + \frac{4r_2 \|p_2\|_{L_1}}{\sqrt{m}} \left[ 1 + \sqrt{\ln(1/\eta)} \right].$$

For the estimates on $|c_1|$ and $\|p_1\|_{L_1}$, note setting $\phi := (2\pi\alpha)^{-1}$, note by Lemma A.3 and a change of variable $w \mapsto \phi w$ and Lemma A.1 that

$$\|p_2\|_{L_1} \leq 4\pi^2 \int |\widehat{f_{|\delta} * G_\alpha}(w)| \int \mathbb{1}[|b| \leq \|w\| \leq r_2] \, db \, dw$$

$$\leq 8\pi^2 \|f_{|\delta}\|_{L_1} \int_{\|w\| \leq r_2} \|w\|(2\pi\phi^2)^{d/2} G_\phi(w) \, dw$$

$$= 8\pi^2 (2\pi)^{d/2} \|f_{|\delta}\|_{L_1} \int_{\|\phi w\| \leq r_2} \|\phi w\| \phi^d G(w) \, dw$$

$$\leq 8\pi^2 (2\pi)^{d/2} \phi^{d+1} \|f_{|\delta}\|_{L_1} \int_{\|\phi w\| \leq r_2} \|w\| G(w) \, dw$$

$$\leq 8\sqrt{d}\pi^2 (2\pi)^{d/2} \phi^{d+1} \|f_{|\delta}\|_{L_1}$$

$$\leq 2\|f_{|\delta}\|_{L_1} \sqrt{\frac{(2\pi)^3 d}{(2\pi\alpha^2)^{d+1}}}.$$

Similarly,

$$|c_2| \leq M + \|f_{|\delta}\|_{L_1} \int (1 + \|w\|)\hat{G}_\alpha(w) \, dw$$

$$\leq M + \|f_{|\delta}\|_{L_1}(2\pi\phi^2)^{d/2} \int (1 + \|\phi w\|) \, dG(w)$$

$$\leq M + 2\sqrt{d}\|f_{|\delta}\|_{L_1}(2\pi\phi^2)^{d/2},$$

$$\|a_2\|_2 = \left\| \int w|\hat{h}(w)| \, dw \right\|$$

$$\leq \int \|w\||\hat{h}(w)| \, dw$$

$$\leq \|f_{|\delta}\|_{L_1}(2\pi\phi^2)^{d/2} \int \|\phi w\||\hat{h}(w)| \, dw$$

$$\leq \sqrt{d}\|f_{|\delta}\|_{L_1}\phi(2\pi\phi^2)^{d/2}.$$

$\square$

# F    DEFERRED PROOFS FROM SECTION 5

Lastly, the two short proofs leading to the RKHS bounds.

*Proof of Proposition 5.1.*    1. By Markov's inequality

$$\int \mathbb{1}[\|\mathcal{T}(\tilde{w})\|_2 > B] \, dG(\tilde{w}) \leq \frac{\|\mathcal{T}\|_{\mathcal{H}}^2}{B^2},$$

thus by Cauchy-Schwarz, for any $\|x\| \le 1$,

$$
\begin{aligned}
\left| \langle \mathcal{T}_B, \Phi(x; \cdot) \rangle_{\mathcal{H}} - \langle \mathcal{T}, \Phi(x; \cdot) \rangle_{\mathcal{H}} \right| &= \left| \int \left( \mathcal{T}_B(\tilde{w}) - \mathcal{T}(\tilde{w}) \right)^{\mathsf{T}} \tilde{x} \sigma'(\langle \tilde{w}, \tilde{x} \rangle) \, \mathrm{d}G(\tilde{w}) \right| \\
&\le \int \left| \mathbb{1}\left[ \|\mathcal{T}(\tilde{w})\|_2 > B \right] \right| \cdot \left| \mathcal{T}(\tilde{w})^{\mathsf{T}} \tilde{x} \right| \cdot \left| \sigma'(\langle \tilde{w}, \tilde{x} \rangle) \right| \mathrm{d}G(\tilde{w}) \\
&\le \sqrt{\int \mathbb{1}\left[ \|\mathcal{T}(\tilde{w})\|_2 > B \right]^2 \mathrm{d}G(\tilde{w})} \cdot \sqrt{\int \left( \mathcal{T}(\tilde{w})^{\mathsf{T}} \tilde{x} \right)^2 \mathrm{d}G(\tilde{w})} \\
&\le \frac{\|\mathcal{T}\|_{\mathcal{H}}}{B} \cdot \sqrt{\int 2 \|\mathcal{T}(\tilde{w})\|^2 \, \mathrm{d}G(\tilde{w})} \\
&= \frac{\|\mathcal{T}\|_{\mathcal{H}}^2 \sqrt{2}}{B}.
\end{aligned}
$$

2. Proceeding similarly, but now using Lemma A.1 to control the indicator,

$$
\begin{aligned}
\left| \langle \mathcal{T}_r, \Phi(x; \cdot) \rangle_{\mathcal{H}} - \langle \mathcal{T}, \Phi(x; \cdot) \rangle_{\mathcal{H}} \right| &= \left| \int \left( \mathcal{T}_r(\tilde{w}) - \mathcal{T}(\tilde{w}) \right)^{\mathsf{T}} \tilde{x} \sigma'(\langle \tilde{w}, \tilde{x} \rangle) \, \mathrm{d}G(\tilde{w}) \right| \\
&\le \int \left| \mathbb{1}\left[ \|\tilde{w}\|_2 > r \right] \right| \cdot \left| \mathcal{T}(\tilde{w})^{\mathsf{T}} \tilde{x} \right| \cdot \left| \sigma'(\langle \tilde{w}, \tilde{x} \rangle) \right| \mathrm{d}G(\tilde{w}) \\
&\le \sqrt{\int \mathbb{1}\left[ \|\tilde{w}\|_2 > r \right]^2 \mathrm{d}G(\tilde{w})} \cdot \sqrt{\int \left( \mathcal{T}(\tilde{w})^{\mathsf{T}} \tilde{x} \right)^2 \mathrm{d}G(\tilde{w})} \\
&\le \sqrt{\exp\left( -(r - \sqrt{d})^2/2 \right)} \cdot \sqrt{\int 2 \|\mathcal{T}(\tilde{w})\|^2 \, \mathrm{d}G(\tilde{w})} \\
&= \frac{\|\mathcal{T}\|_{\mathcal{H}} \sqrt{2}}{e^{(r - \sqrt{d})^2/4}}.
\end{aligned}
$$

$\square$

