# OpenReview forum: "Neural tangent kernels, transportation mappings, and universal approximation"
_ICLR.cc/2020/Conference — Accept (Poster)_

### Official Review · AnonReviewer3 · 2019-10-22
**Official Blind Review #3**

**Rating:** 6

**Review:**

The paper studies approximation properties (in L2 over some data distribution P) of two-layer ReLU networks in the NTK setting, that is, where weights remain close to initialization and the model behaves like a kernel method given by its linearization around initialization.

The authors obtain a variety of results in order to obtain such approximation guarantees, which are obtained by sampling from a so-called 'transport mapping', which is essentially a function T:R^{d+1}->R^{d+1} with a bound on sup_w ||T(w)||, which can approximate well various classes of target functions (section 3).
In particular, they show that such a sampling leads to weights close to initialization, that the neural network function is close to its linearization in L2(P), and that the linearization is close to the target function in L2(P).
Together with a control of the norm of T required to approximate the target function, this leads to approximation bounds in Theorem 1.3.

The techniques used to obtain transport mappings are quite interesting and seem novel, and the general approach for controlling various steps from neural network function to the target function in L2(P) norm in the NTK setting is insightful and novel as far as I know.
That said, the presentation lacks a certain amount of polish in its present form, which makes me lean towards the reject side. I also have some comments related to novelty of certain aspects.

Comments:
* the paper is not well organized, with the main result appearing in the introduction with little details on the involved quantities, no clear separation or connection between intermediate lemmas in later sections, and very little motivation and explanation of some results. Further, there are many typos and inconsistent notations throughout which make the paper hard to read.

* the sampling result in Lemma 2.1 is very similar in flavor to random feature approximation results (for the NTK here), e.g., [1, Proposition 1], which could perhaps be more precise in practice as it is data-dependent, and only needs an L2 control on the function T. Can this be applied here or would the initialization term mess things up? A comparison would be helpful either way.

* the approximation rates should be discussed more (are they optimal?), and compared to prior work, both on general two-layer networks, and kernels arising from a similar setup, in particular in [2] and the cited Sun et al. (note that the NTK behaves similarly in terms of approximation, see [3])

* the section on the "natural RKHS" is largely unclear, as is the corresponding bound in Theorem 1.3 (shouldn't B be proportional to the RKHS norm?)

* how these results apply to networks obtained via optimization in the NTK regime should probably be discussed more

smaller things:
* eq (1.2): what is the mearning of the epsilon factor? is it standard?
* p.2 "to not yield" -> do not yield
* "with scaling... width": rephrase (and, do you mean dataset size?)
* "one 1/m is then pushed" -> one 1/sqrt(m)?
* throughout: pick a consistent notation for derivative of relu (sometimes it's sigma', sometimes an indicator)
* section 2: here it seems like T(w)/(eps sqrt(m)) is just the movement from initialization and T_m,eps(w) are the final weights, while in the introduction T(w) indicates the final weights, this is confusing notation. Also, shouldn't T_m,eps appear in the bounds?
* section 3: should -> shows?
* lemma 3.1, 3.2: specify that the other coordinates are 0, also missing dG(w) in the definition of g. What do we lose from the use of truncation?
* lemma 3.5: sup_x or just L2(P)?
* section 3.3: H is just L2(G) here? Also, the kernel is not universal with only even terms, but the bias fixes that, see e.g. [2,4].


[1] Bach. On the Equivalence between Kernel Quadrature Rules and Random Feature Expansions (2017)
[2] Bach. Breaking the Curse of Dimensionality with Convex Neural Networks (2017)
[3] Bietti and Mairal. On the Inductive Bias of Neural Tangent Kernels (2019)
[4] Basri et al. The Convergence Rate of Neural Networks for Learned Functions of Different Frequencies (2019)


===== update post rebuttal =====

Thanks for the detailed response, I am increasing my score as the new version looks much better.

I am a bit puzzled (and surprised) by the gap in the rates between NTK and relu random features, as it seems to suggest that only training second-layer weights while leaving the first layer at random initialization yields better rates than the NTK regime, if I understand correctly? If so, is this due mainly to the linearization step, i.e. Lemma 2.6? It would be good to include some further discussion on this in the paper.

As per approximation by random features/sampling, note that [1, Prop. 1] only requires a sup control on random features (which is quite trivial here with bounded data), not on the "transport" (beta in their statement).

**Experience Assessment:**

I have published one or two papers in this area.

**Review Assessment: Checking Correctness Of Derivations And Theory:**

I assessed the sensibility of the derivations and theory.

**Review Assessment: Checking Correctness Of Experiments:**

N/A

**Review Assessment: Thoroughness In Paper Reading:**

I read the paper at least twice and used my best judgement in assessing the paper.

---

> ### Author Response · Authors · 2019-11-15
> **Response to AnonReviewer3.**
>
> We thank the reviewer for their thorough comments.
>
> What we wish to primarily highlight is that we have heavily restructured
> and polished the paper.  As the paper only has two reviews, we hope the
> reviewer is able to find the time to at least go through the new
> abstract and introduction.  We have summarized the changes in a separate
> comment above, but a list of notable changes which match the comments of
> the reviewer are as follows:
>
> - In response to "the main result appearing in the introduction with
>   little details on the involved quantities", we have (a) pushed the
>   main result back to page 3, (b) pushed before it the key idea,
>   originally in section 2, that one can easily take a network written
>   with a transport and obtain another (wider) network where weights are
>   close to initialization, (c) focused the main theorem and introduction
>   on the approximation of continuous functions, with all relevant
>   quantities defined within the theorem or before, (d) moved the other
>   parts of the theorem later, and better modularized and pre-empted
>   them.
>
> - Regarding "no clear separation or connection between intermediate
>   lemmas in later sections, and very little motivation and explanation
>   of some results", after moving parts of the main theorem out and
>   pushing them to other sections, we then better modularized the
>   different results and sections, and included further explanations.  It
>   is for this reason that the paper is now longer.  As a concrete
>   example, Section 2, on sampling, now starts with a single main theorem
>   encapsulating the technique; before, this theorem was split into
>   parts, and much more unwieldy to apply.
>
> - Regarding "many typos and inconsistent notations", we have performed
>   extensive editing.
>
> We hope the reviewer finds the presentation vastly improved, and
> appreciate further comments.

---

> > ### Author Response · Authors · 2019-11-15
> > **(Continuation of response to AnonReviewer3.)**
> >
> > In response to other comments:
> >
> > - Regarding the sampling tool in the reviewer's reference [1,
> >   Proposition 1], in fact the dependence is nearly identical to ours:
> >   our proof also gives a supremum over the "basis" (in this case
> >   depending on the transport T) and an L_2 dependence on the data
> >   measure; for simplicity we've enforced ||x|| <= 1 and hidden this L_2
> >   as it is generally small compared to the dependence on T.  We further
> >   note that these sampling tradeoffs are also similar to those in [2,
> >   Proposition 1], cited by the reviewer (and by the same author).
> >
> > - Regarding optimality of our rates, we have included concrete
> >   discussions of lower bounds, one from the reviewer's reference [2],
> >   and another from a paper by Yarotsky in our revisions.  Concretely,
> >   both of these reference suggest lower bounds of the form 1/eps^{d/2}
> >   if weights are not required to be close to initialization, where in
> >   our case our upper bounds are closer to 1/eps^{10d} while lying close
> >   to initialization.  We further note that the lower bound in [2] has
> >   further restrictions (e.g., data on the sphere), and Yarotsky's paper
> >   also has a higher lower bound 1/eps^{d} when the constructions are
> >   "continuous", which we discuss in our revisions.  The upper bounds
> >   (not in the NTK setting) presented in [2] are roughly tight with the
> >   lower bounds, though as we mentioned the setting there is uniform on
> >   the sphere.  We also note that our tools here (giving 1/eps^{10d} in
> >   the NTK setting) give a much better 1/eps^{2d} when applied to regular
> >   networks with a single hidden layer, as now included in section 4.
> >
> > - Despite pushing the RKHS material farther back, we have expanded it,
> >   and hope it is clearer now.  We have removed the universal
> >   approximation comments, since they are contained in the work by Sun et
> >   al.
> >
> > - Regarding the NTK optimization literature, we hope that the present
> >   results can be helpful in proving good test error bounds.  As a
> >   concrete example, a work we cite by Arora et al generalizes well if
> >   the quantity y^T (H^\infty)^{-1} y is small, which is saying that the
> >   features and labels share some structure.  Writing the optimal labels
> >   as function of the inputs (i.e., the least squares solution over the
> >   population), one can now introduce our tools, and hopefully develop
> >   more concrete rates; we have included a version of this comment
> >   in the concluding open problems section.
> >
> > - The epsilon factor in the NTK definition appears in a variety of
> >   works, and corresponds to the initialization of the second layer
> >   weights; see for instance the work by [ Allen-Zhu, Li, Liang ] which
> >   we cite.  This scaling factor appears crucial and we highlight it
> >   better in our new version (e.g., see the blue bolded terms on pages
> >   3 and 4).
> >
> > - We have standardized our notation in the body to write sigma' in place
> >   of an indicator, though on page 2 we write the indicator once for sake
> >   of concreteness, and also write it in some proofs.
> >
> > - The new lemmas 3.2 and 3.3 clearly fix the first d coordinates as 0.
> >   The second part of lemma 3.1, as we discuss in section 5, can be used
> >   to form a mapping that uses all coordinates, but it does not seem to give
> >   better bounds (after all it is similarly just a rescaling).  Regarding
> >   what is lost with truncation, unfortunately it is unclear.  To
> >   highlight where the inefficiencies in our techniques may lie, we have
> >   included Theorem 4.5, which uses our techniques to prove a width upper
> >   bound on approximating continuous functions with random features, and
> >   it improves the earlier 1/eps^{10d} to 1/eps^{2d}, and uses no
> >   truncations.  We are not asserting here that the truncations lead to
> >   the extra factors, however certainly they make the proofs and bounds
> >   significantly messier, and correspondingly this makes it harder to
> >   produce tight bounds.
> >
> > - The "B" for the RKHS transports indeed depends on the RKHS norm; the
> >   earlier attempt at rushing to the main theorem within two pages made
> >   this unclear, but it should now be clear, with two separate bounds
> >   now appearing in section 5, both depending on the RKHS norm.
> >
> > - The Hilbert space we develop to discuss RKHSes is indeed the same as
> >   L2(G); we have included this clarification, thank you.
> >
> > - Thank you for the four references, we have included them.

---

> ### Author Response · Authors · 2020-02-15
> **Thank you for the detailed post-rebuttal comments**
>
> Thank you for evaluating our revised paper; it was a lot of work for you, given our extensive changes.  We thank you for your very thorough and valuable comments.
>
> In response to your post-rebuttal comments, we've updated our "open problems" section to expand on these gaps.
>
> To respond informally to you here, I agree, it is odd.  Of course, the "10" we have must be an an analytic artifact, however I don't know what it should be.  In that new open problem comment (which is admittedly quite brief), I highlight both the choice you mention (which layers do you train), but also the question of norm (our paper here is in the "NTK standard" (2,infty) norm).   Would be nice to know all the gaps, which choices are relevant in practice, how they affect optimization and generalization, how depth changes things, ...
>
> Thanks again!

---

### Official Review · AnonReviewer2 · 2019-10-28
**Official Blind Review #2**

**Rating:** 8

**Review:**

Summary: the paper consider representational aspects of neural tangent kernels (NTKs). More precisely, recent literature on overparametrized neural networks has identified NTKs as a way to characterize the behavior of gradient descent on wide neural networks as fitting these types of kernels. This paper focuses on the representational aspect: namely that functions of appropriate "complexity" can be written as an NTK with parameters close to initialization (comparably close to what results on gradient descent get).
The main technical ingredients are a constructing a "transport" map via a Fourier-expansion style averaging (ala Baron), and subsequently subsampling this average ala Maurey-style analyses to get a finite width average.
The authors also identify function classes which are well-behaved with respect to these techniques: smoothed functions (via convolving with a Gaussian), functions which have a small RKHS norm (for an appropriate RKHS derived from NTKs), functions with small modulus of continuity.

Evaluation: the paper is a strong contribution, on a topic which is of great current interest, and I recommend acceptance. It is very nice that many of the standard tools in approximation theory (Fourier expansions, Maurey sampling, etc.) play nicely with NTKs, and also that the scaling of the # of neurons necessary that appears in the current literature can be also recovered via a representation theoretic viewpoint. The paper is written well, and is easy to read.

Minor comments:
* I'd rearrange the bullets bounding B_{f,\epsilon} for the various subcases of Theorem 1.3: I think the RKHS is the most "vanilla" bound, given that you can extract a RKHS; bounds in terms of the modulus of continuity should go next (this is the "weakest" assumption); smoothed f's should go last (this is like a smoothed complexity kind of result)
* w_f isn't defined until section 3.2 -- I'd put a pointer in the statement of Theorem 1.3 to the equation, not just the section.
* I'm not sure "transport" is the ideal term -- it brings to mind "optimal transport", and I kept expecting some Wasserstein connection.



**Experience Assessment:**

I have published in this field for several years.

**Review Assessment: Checking Correctness Of Derivations And Theory:**

I assessed the sensibility of the derivations and theory.

**Review Assessment: Checking Correctness Of Experiments:**

N/A

**Review Assessment: Thoroughness In Paper Reading:**

N/A

---

> ### Author Response · Authors · 2019-11-15
> **Response to AnonReviewer2.**
>
> We thank the reviewer for their comments and support.
>
> In particular, we are grateful for the comment that the "paper is
> written well, and easy to read", and hope it does not seem to odd that
> we went through a significant restructuring.  We hope the paper has only
> more of what AnonReviewer2 liked, not less.
>
> Regarding specific comments:
>
> - Rather than merely re-ordering the main theorem, we have simplified it
>   as discussed in our revision summary, moving all but the continuous
>   function approximation to other sections.
>
> - We agree that the title is suggestive of optimal transport, and indeed
>   worked both before and after the deadline to develop such a
>   connection.  Unfortunately, it seems somewhat elusive to derive
>   anything concrete, and have included a brief remark in the open
>   problem section; thank you for highlighting this.

---

### Author Response · Authors · 2019-11-15
**List of revisions.**

We have restructured the paper and improved presentation, thanks to
helpful feedback from AnonReviewer3.

A summary of the major re-arrangements is as follows.

- In order for the main theorem to be more digestible, it has been pared
  down to only discuss continuous functions, and its other components
  have been pushed to later sections.

- To further aid in the exposition of the main theorem, more
  explanations have been moved before it; notably the central
  description of the sampling method, and how it gives rise to the NTK
  setting of small weight changes, has been moved from section 2 to page
  2 of the introduction.

- The remaining sections have been made more modular, and there are now
  six sections in place of four.

- In a bit more detail: section 2 still contains the old sampling
  routines, though with re-organized exposition, and a single main
  sampling theorem at the start encapsulating the technique; section 3
  now contains only the Fourier-based transportation mappings; section 4
  approximates continuous functions; section 5 briefly describes
  "abstract" mappings, including the old RKHS-based transports; section
  6 concludes with open problems.

(Note that uploading of new abstracts is seemingly disabled; the revision
has a new abstract corresponding to the newly focused presentation.)

In order to flesh out the story and aid exposition (but not changing the core
results and specifically not trying to burden reviewers), a few new extensions
have been added:

- The tools of the paper have been applied directly, in Theorem 4.5, to
  approximation via regular shallow networks (and not the NTK); the
  approximation rates improve, revealing a tantalizing direction for
  future work.

- Appendix B summarizes the low level sampling routines, and now
  includes uniform norm sampling tools, which are applied in the proof
  of Theorem 4.5 as a demonstration; in particular, L_2(P) was not
  essential.

The revisions also include numerous other typo fixes, missing
references, and other corrections, with many thanks to the reviewers.

---

### Decision · Program_Chairs · 2019-12-19

**Decision:**

Accept (Poster)

**Comment:**

The paper considers representational aspects of neural tangent kernels (NTKs). More precisely, recent literature on overparametrized neural networks has identified NTKs as a way to characterize the behavior of gradient descent on wide neural networks as fitting these types of kernels. This paper focuses on the representational aspect: namely that functions of appropriate "complexity" can be written as an NTK with parameters close to initialization (comparably close to what results on gradient descent get).

The reviewers agree this content is of general interest to the community and with the proposed revisions there is general agreement that the paper has merits to recommend acceptance.